# Cyclin *CLB2* mRNA localization and protein synthesis link cell cycle progression to bud growth

Anna Maekiniemi [1,7], Philipp Savakis[2,7], Kelly van Rossum [2], Jacky L. Snoep [3,4], Markus Seiler[5], David D. van Niekerk [3], Kathi Zarnack [5,6], Robert H. Singer [1] & Evelina Tutucci [1,2] ✉

Clb2 is a conserved B-type cyclin essential for mitotic progression in *Saccharomyces cerevisiae*, with expression tightly regulated at transcriptional and proteolytic levels. However, it remains unclear whether Clb2 protein synthesis is regulated and responsive to cell growth. Here, we show that *CLB2* mRNA localizes to the bud via the She2/She3 complex, while Clb2 protein accumulates in the mother nucleus. This mRNA localization enhances translation without affecting protein localization. A structured RNA element, a ZIP-code, is located within the coding sequence and is required, but not sufficient, for both mRNA transport and protein expression. Mutation of this ZIP code disrupts mRNA localization, reduces Clb2 synthesis, increases budded phase duration and daughter cell size. In wild-type cells, Clb2 protein levels scale with bud growth, a coupling lost in ZIP code mutants. These findings reveal a mechanism by which mRNA localization and translation are coordinated to link cell growth with cell cycle progression.

Over the past decades, RNA imaging technologies have revealed that hundreds of mRNAs localize to various subcellular compartments, in organisms ranging from bacteria to multicellular eukaryotes[1,2], suggesting that mRNA trafficking is a conserved and integral part of gene expression regulation. Current studies suggest that the primary role of mRNA trafficking is to control asymmetric protein distribution to sustain local functions such as cell migration and polarity[1]. Even in the single-cell organism *S. cerevisiae*, dozens of mRNAs localize to the endoplasmic reticulum, mitochondria, and the bud[3]. The best-characterized localized mRNA is *ASH1*, which is transported to the yeast bud on actin filaments by the She2-She3 complex and the type V myosin motor Myo4[4–11]. The RNA-binding proteins (RBP) Khd1 and Puf6 bind the *ASH1* mRNA and inhibit its translation until the bud-localized kinases Yck1 and CK2 phosphorylate Khd1 and Puf6, releasing

the inhibition and allowing local translation to occur[12–17]. The Ash1 protein is subsequently asymmetrically segregated into the daughter nucleus, where it controls mating-type switching[3,7]. An additional kinase-RBP pair, Cbk1-Ssd1, has been shown to localize to the bud[18] and tune the translation of specific mRNAs[19–21].

Besides *ASH1*, multiple mRNAs have been shown to interact with the She2-She3-Myo4 complex and to localize to the bud[22]. Among these mRNAs is *CLB2*, which encodes a conserved B-type cyclin, interacting with and controlling the substrate specificity of the cyclin-dependent kinase Cdk1[23–28]. The Clb2 protein contains two nuclear localization signals and two nuclear export signals[29]. Most of the protein is found in the nucleus and at spindle pole bodies[29–31]. However, when the protein is overexpressed or in nuclear import mutants, the Clb2 protein accumulates in the cytoplasm or at the bud neck[29–31].

[1]Cell Biology, Albert Einstein College of Medicine, 1300 Morris Park Avenue, Bronx, NY, USA. [2]A-LIFE Department, Systems Biology Section, Amsterdam Institute of Molecular and Life Sciences (AIMMS), Vrije Universiteit Amsterdam, De Boelelaan 1108, Amsterdam, The Netherlands. [3]Department of Biochemistry, University of Stellenbosch, Stellenbosch, South Africa. [4]Department of Molecular Cell Biology, Vrije Universiteit Amsterdam, De Boelelaan 1085, Amsterdam, The Netherlands. [5]Buchmann Institute for Molecular Life Sciences (BMLS) & Faculty of Biological Sciences, Goethe University Frankfurt, Max-von-Laue-Str. 15, Frankfurt, Germany. [6]Theodor Boveri Institute, Julius Maximilians University Würzburg, Biocenter, Am Hubland, Würzburg, Germany. [7]These authors contributed equally: Anna Maekiniemi, Philipp Savakis. ✉e-mail: evelina.tutucci@vu.nl

Clb2-Cdk1 regulates entry and progression through mitosis in a threshold-dependent manner[25,32–36], by phosphorylating transcriptional and post-transcriptional regulators[37–39]. This triggers a positive feedback loop leading to the transcription of the CLB2 cluster, a set of 35 genes including *CLB2*, expressed during the G2/M phase transition[24,40–42]. Aberrant Clb2 expression -depletion or over-expression- results in abnormal mitotic progression and cell size alteration[25,27,43]. To achieve accurate periodic Clb2 expression, cells combine cell-cycle-dependent mRNA synthesis[44,45], controlled mRNA decay[46], and proteasome-dependent protein degradation[47,48]. The molecular events controlling *CLB2* transcription and protein degradation, as well as Clb2 function during cell cycle progression, are well characterized. However, it remains unclear whether and how cells modulate *CLB2* mRNA translation in response to changes in cell growth, such as bud growth during *S. cerevisiae* mitosis, and how it may be coupled to cell cycle progression. Previous mathematical modeling suggested that *CLB2* mRNA localization and local translation could act as a bud sizer during the G2/M phase checkpoint[49], but thus far experimental evidence supporting this hypothesis is lacking.

To study growth phase-dependent *CLB2* expression regulation, we performed single-cell measurements of *CLB2* mRNA and protein expression throughout the budding yeast cell cycle in fixed and living cells. We combined single-molecule RNA fluorescence in situ hybridization (smFISH)[50,51] and immunofluorescence (IF)[52,53] to detect *CLB2* mRNA and its protein product simultaneously in individual cells. Furthermore, to study dynamic gene expression changes in living cells, we utilized the MS2 system (MBSV6) optimized to endogenously tag unstable mRNAs in *S. cerevisiae*[54–57], and tagged the Clb2 protein with GFP. Our work shows that *CLB2* mRNAs are efficiently localized in the bud by the She2-She3 complex during the G2/M phase, while the CLB2 protein is found in the mother nucleus. We show that the *CLB2* mRNA has a single ZIP code located in the coding sequence, which is necessary but not sufficient for both mRNA localization and Clb2 protein synthesis. We found that bud localization stimulated *CLB2* mRNA translation, as in a *CLB2* ZIP code mutant, we observed impaired mRNA bud localization and reduced Clb2 protein synthesis, which were partially rescued in ZIP code rescue strains. Furthermore, we found that the ZIP code mutant showed an increase in bud size at birth and an increase in budded phase duration. Finally, unlike in wild-type (WT) cells, we observed that protein accumulation in the ZIP code mutant showed reduced correlation with bud growth and thus lost the ability to be a consistent predictor of size changes occurring in this compartment. Altogether, we propose that yeast cells have evolved a mechanism that couples the control of *CLB2* mRNA bud localization and protein synthesis to regulate Clb2 protein levels in response to bud growth. We suggest that, in coordination with other known mechanisms controlling *CLB2* expression, this mechanism helps cells monitoring bud growth and fine-tune cell cycle progression.

## Results

### *CLB2* mRNAs localize in the bud from S phase to Mitosis
To quantify *CLB2* mRNA expression throughout the *S. cerevisiae* cell cycle, we combined smFISH and IF. To monitor cell cycle progression, nuclear localization of the transcription factor Whi5 was used to classify early G1 phase, while G2 and mitotic cells were identified by staining tubulin (Tub1) and monitoring microtubules stretching between the mother and the daughter mitotic spindles (Fig. 1a). *CLB2* smFISH revealed that mRNAs are detected from late S phase, when the bud emerges from the mother cell, until the end of anaphase. Quantification of *CLB2* mRNAs showed that mRNAs are found in 63% of cells in an unsynchronized population (Fig. 1b, Supplementary Fig. 1a). The expression peak occurred during G2 (average $10.2 \pm 5.7$ mRNAs/cell) when about 50% of the cells showed an active transcription site (Fig. 1b, c, yellow arrows) with on average $2.9 \pm 1.5$ nascent RNAs per

transcription site, comparable to previous studies[46] (Supplementary Fig. 1b). Furthermore, in expressing cells, *CLB2* nascent RNA expression showed Poissonian transcription kinetics typical of constitutive genes[58], suggesting that this cell-cycle regulated gene is likely transcribed in a single activation event with fixed initiation rate. From late S phase until anaphase, we observed that *CLB2* mRNAs localize to the bud from the first stages of bud formation. Throughout the budded phases, we measured up to 65.6% of mRNAs in the bud, as compared to the distribution of the non-localized mRNA *MDN1*, where only 17.2% of mRNAs are found in this compartment (Fig. 1b, d, Supplementary Fig. 1c, d). *CLB2* mRNA bud localization is independent of the *S. cerevisiae* background since we observed it both in BY4741, used throughout this study, as well as in the W303 background (Supplementary Fig. 1e).

### *CLB2* mRNAs localize in the bud of living *S. cerevisiae* cells
To investigate *CLB2* mRNA localization dynamics in living cells, we used an MS2 system optimized for yeast mRNA tagging (MS2 binding sites V6, MBSV6)[54–56]. We inserted 24xMBSV6 in the 3'UTR (Untranslated region) of the endogenous *CLB2* locus (Supplementary Fig. 2a). To confirm that mRNA tagging with MBSV6 did not alter *CLB2* mRNA expression, unlike with previous MS2 variants[55,59,60], we performed two-color smFISH with probes targeting either the coding sequence (CDS) or the MBSV6 loops, to compare the expression of the endogenous and the tagged *CLB2* mRNA. This confirmed that MS2-tagged mRNAs are full-length and correctly localized in the bud (Supplementary Fig. 2b). Furthermore, comparable mRNA levels were observed whether the mRNA was MS2-tagged, with or without GFP-tagged MS2 coat protein (MCP-GFP), which is used to detect mRNAs in living cells (Supplementary Fig. 2c–e).

To monitor cell cycle progression and bud emergence in living cells, we tagged the bud neck protein Cdc10 endogenously with tdTomato in the *CLB2*-MS2-tagged strain (Supplementary Fig. 3a). We performed time-lapse imaging every 2 min and measured *CLB2* mRNA expression throughout the cell cycle by acquiring z-stacks encompassing the cell volume (Video 1). To reduce perturbations in gene expression due to synchronization protocols[61], we quantified *CLB2* mRNA expression in unsynchronized cells, using the bud neck marker expression to compare cells. This revealed that up to 62.9% of *CLB2* mRNAs localized in the bud (Fig. 1e, f), in agreement with the smFISH quantifications (Fig. 1d). Furthermore, mRNAs were degraded before the end of mitosis with a half-life of $3.8 \pm 1.4$ min, consistent with previous measurements[46], suggesting that the MS2 system does not affect *CLB2* mRNA stability (Supplementary Fig. 3b). Interestingly, imaging of mother-daughter pairs for more than one cell cycle showed that the daughter cell initiated *CLB2* mRNA expression about 20 min after the mother (Fig. 1g). This observation is in line with previous evidence showing that *S. cerevisiae* daughter cells are born significantly smaller than mothers and that cell size control occurring during G1 regulates the entry into the next cell cycle[62–64].

High framerate imaging every 100 milliseconds (ms) revealed that, as the bud grows, the number of *CLB2* mRNAs localized in the bud rapidly increases (Supplementary Fig. 3c, Video 2). The high concentration of mRNAs in the bud and their rapid movement in and outside of the focal plane did not allow tracking single mRNA molecules to investigate *CLB2* mRNA bud localization dynamics. We therefore used mathematical modeling to predict the behavior of *CLB2* mRNA in the bud of an average G2 cell. The parameters included the bud and mother volumes based on our measurements, mRNA counts and decay rates based on smFISH and live imaging experiments and an apparent mRNA diffusion coefficient based on previous measurements performed in eukaryotic cells[65] ("Methods"). A fast coefficient of $0.4 \, \mu m^2/s$ was previously measured for non-translating mRNAs, and a slower coefficient of $0.1 \, \mu m^2/s$ was measured for translating mRNAs[65]. Interestingly, our model

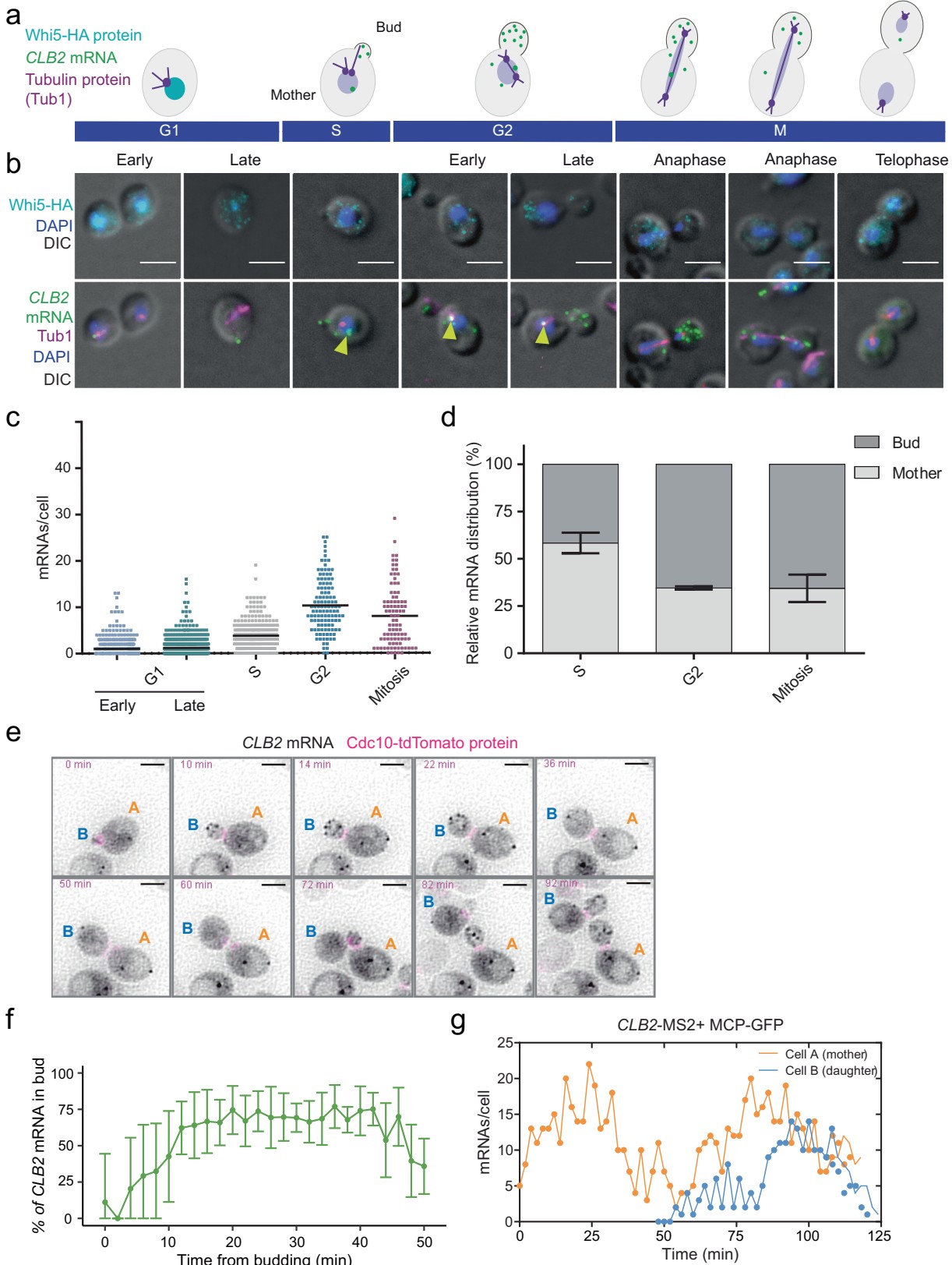

suggested that if we assumed either a slow or a fast apparent mRNA diffusion coefficient of 0.1 μm²/s or 0.4 μm²/s, respectively, we would not obtain the expected enrichment of the *CLB2* mRNA in the bud (Supplementary Fig. 3d, e). To predict the accumulation of about 65% of the mRNA in the bud observed during the G2 phase, the presence of a high-affinity anchoring factor promoting *CLB2*

mRNA segregation in the bud had to be included in the simulation (Supplementary Fig. 3f). Altogether, these results showed that *CLB2* mRNAs were efficiently transported to the bud in a cell cycle dependent manner and the simulation implied that the mRNA might be anchored via a yet uncharacterized mechanism.

**Fig. 1 | *CLB2* mRNAs localize to the bud in a cell-cycle dependent manner.**
**a** Schematic of *CLB2* mRNA expression during the cell cycle. Green dots represent *CLB2* mRNAs. The Whi5 protein (cyan) accumulates in the nucleus during early G1. Tubulin (magenta) is a component of microtubules and the mitotic spindle. The bud stage starts during the S phase and ends with the formation of the daughter cell. During anaphase, microtubules stretch between the mother and the daughter cell. **b** Top panels: MERGE Maximal projections of IF anti-HA (Whi5) (cyan) and DAPI (blue) merged to a single differential interference contrast (DIC) section (gray). Bottom panels: MERGE Maximal projections of *CLB2* mRNA smFISH (green), anti-tubulin IF (magenta) and DAPI (blue) merged to a single DIC section (gray). The corresponding cell cycle phase is indicated on the panels. Scale bars 3 μm. **c** smFISH quantifications of *CLB2* mRNA expression during the different cell cycle phases determined using the markers shown in (**b**). Dots correspond to individual cells (2083 cells, pooled from two biological replicates). The black bar indicates the mean (G1 early = 1.2 ± 1.9; G1 late = 1.2 ± 1.9; S = 3.8 ± 3.0; G2 = 10.3 ± 5.7; M = 8.0 ± 6.5 mRNAs/cell, mean ± SD). **d** Relative distribution (bud vs mother) of *CLB2* mRNA in WT budded cells based on the smFISH-IF data shown in (**b**) (S phase = 58.4 ± 5.5, G2 phase = 34.6 ± 0.9, M phase = 34.4 ± 7.2; mean ± SD of cells pooled from two biological replicates). **e** *CLB2* endogenously tagged with 24 × MBSV6 to enable visualization of the mRNA (black) in live cells. The bud neck protein Cdc10-tdTomato is shown in magenta. Cell A is the mother of cell B. Time point from the start of acquisition is indicated in the upper left corner of each time frame. Scale bars 3 μm. **f** Percentage of *CLB2* mRNAs localized in the bud over time from bud appearance. Error bars indicate mean ± SD of 9 cells pooled from 3 biological replicates. **g** Number of *CLB2* mRNAs per cell over time in cell A and cell B, shown in panel (**e**). Representative analysis of cells collected over three biological replicates. Source data are provided as a Source Data file.

## She2 and She3 are required for ZIP-code-dependent *CLB2* mRNA localization

To elucidate the function of *CLB2* mRNA localization, we first characterized the mechanism of *CLB2* mRNA transport. Previous work showed that *CLB2* mRNA is associated with the She2-She3-Myo4 complex and that localization of *CLB2* to the bud likely requires She2[22]. To expand this observation and quantify this phenotype, we performed smFISH-IF throughout the cell cycle for the *CLB2* mRNA in *SHE2* or *SHE3* gene deletion strains to test whether the She2-She3 complex, required for *ASH1* mRNA transport[4–11], was also involved in *CLB2* mRNA localization. This revealed that in *Δshe2* and *Δshe3* strains, localization was strongly impaired (Fig. 2a, Supplementary Fig. 4a) and that during mitosis, only up to 24.5% and 23.6% of *CLB2* mRNAs were found in the bud of the *Δshe2* and *Δshe3* strains, respectively (Supplementary Fig. 4b, c).

As the She2-She3 complex was required for *CLB2* mRNA localization, we hypothesized that the *CLB2* mRNA might possess a ZIP code akin to the *ASH1* ZIP code. Previous work defined the sequence and structure of the (E3) *ASH1* mRNA ZIP code bound by She2[8,66–69]. Based on sequence and structure similarity, a pattern search was performed to predict occurrences within the *CLB2* mRNA ("Methods"). We identified one high-confidence She2-binding site in the CDS at position 1111–1145 (Fig. 2b, c). To test the predicted site, we generated a *CLB2* synonymized mutant whereby the CDS was mutagenized at nine bases to destroy the ZIP code structure, while preserving the protein sequence and the codon usage index (*ZIP mut*, Fig. 2d, Supplementary Fig. 4d, e). A pattern search confirmed that the ZIP code was destroyed upon synonymization. smFISH revealed that, in the *CLB2* ZIP code mutant, the *CLB2* mRNA was no longer bud localized (Fig. 2e, f), demonstrating that the identified ZIP code is necessary to control bud mRNA localization, possibly by recruiting the She proteins. Furthermore, the loss of *CLB2* mRNA localization was not due to the specific mutations introduced, as other mutants with disrupted ZIP code structure also showed *CLB2* mRNA mis-localization (Supplementary Fig. 4d, f). To measure *CLB2* mRNA localization, we quantified the mRNA peripheral distribution index (PDI) in budded cells using the RNA Distribution Index Calculator[70] ("Methods"). The PDI measures the location of the mRNA relative to the nucleus and allows for the comparison of the localization of distinct mRNA species. An index value equals 1 for diffusely distributed mRNAs or >1 if the mRNA has a polarized pattern, and <1 if the mRNA is distributed closer to the nucleus[70] (Fig. 2g). This analysis revealed a PDI of 1.9 ± 0.42 for the *CLB2* mRNA, similar to the index value of the control mRNA *ASH1* (PDI = 2.2 ± 0.43, Fig. 2h). The PDI value was significantly reduced for *CLB2* in the *Δshe2* (PDI = 0.5 ± 0.2), *Δshe3* (PDI = 0.4 ± 0.13), and *CLB2* ZIP code mutant strain (PDI = 0.5 ± 0.16) (ANOVA statistical test: $F_{(4, 185)}=15.74$, $p<0.0001$), with PDI values similar to the non-localized mRNA *MDN1* (PDI = 0.6 ± 0.18, Fig. 2h, Supplementary Fig. 1c, d). Thus, the She2–3 complex is required to transport *CLB2* mRNAs to the bud via a conserved ZIP code sequence.

## *CLB2* mRNA ZIP code and bud localization regulate Clb2 protein levels

To elucidate whether *CLB2* mRNA localization influences gene expression, we measured *CLB2* mRNA and protein levels in the localization mutants. Using smFISH, we found comparable mature and nascent RNAs counts in the *Δshe2* (5.3 ± 5.1 mRNA/cell and 2.7 ± 1.1 nascent RNA/transcription site (TS)) or *Δshe3* mutants (5.2 ± 6.2 mRNA/cell and 2.7 ± 1.3 nascent RNA/TS) compared to WT cells (5.4 ± 5.1 mRNA/cell and 2.6 ± 1.3 nascent RNA/TS). While we observed a small but significant increase in the *CLB2* ZIP code mutant strain (6.9 ± 6.3 mRNA/cell and 3.4 ± 1.3 nascent RNA/TS) compared to WT cells (Fig. 3a, b). Next, a western blot of the endogenously myc-tagged Clb2 protein showed that the protein expression in *Δshe2* or *Δshe3* and in the *CLB2* ZIP code mutants was significantly reduced compared to WT cells (Fig. 3c, d). Because the ZIP code mutant showed a stronger decrease in Clb2 expression compared to the *Δshe* mutants, we tested whether the *ZIP*[2] and *ZIP*[3] mutants showed a similar phenotype (Supplementary Fig. 4d, e, Supplementary Fig. 5a, b). As all the mutants showed the same reduction in Clb2 protein expression, this indicated that this region of the *CLB2* mRNA is important to control mRNA localization and protein expression. In view of the similarity in the mRNA localization and protein expression phenotypes of the three ZIP mutant strains, we continued the characterization of ZIP code mutant 1 (ZIP[1]), henceforth called 'ZIP-mutant'. Furthermore, to test whether the effect of the She proteins and the ZIP code on protein expression was independent of each other, we generated *Δshe2*-ZIP or *Δshe3*-ZIP double mutants (Supplementary Fig. 5c–e). Protein expression analysis by western blot showed no significant decrease in protein expression in the double mutants compared to the ZIP mutant, and smFISH showed a lack of mRNA localization in the bud. Altogether, revealing an epistatic interaction and thus suggesting that the She proteins and the ZIP code of the *CLB2* mRNA act in the same pathway.

To test whether the decrease in protein expression was due to a change in protein degradation, we performed stability assays by treating WT and localization mutants with the translation inhibitor cycloheximide and measured the protein abundance over time (Fig. 3e, f, Supplementary Fig. 5f, g). No significant difference was observed in the stability of the localization mutants compared to WT cells, suggesting that *CLB2* ZIP code and mRNA localization, rather than mRNA or protein stability, regulated Clb2 protein synthesis.

Finally, we investigated whether mRNA localization affected Clb2 protein localization. To this end, *CLB2* was endogenously tagged with yeast-optimized GFP (yeGFP) in WT and *CLB2* mRNA localization mutants. We observed that Clb2 is predominantly found in the nucleus in all tested strains (Fig. 3g), as previously reported for WT cells[29–31]. Interestingly, Clb2 was observed in the mother nucleus already during the G2 phase (Fig. 3g, top panels), when the mRNAs were already localized to the bud (Figs. 1b, e, 2a), showing an uncoupling between *CLB2* mRNA and protein localization. Altogether, these results suggested that

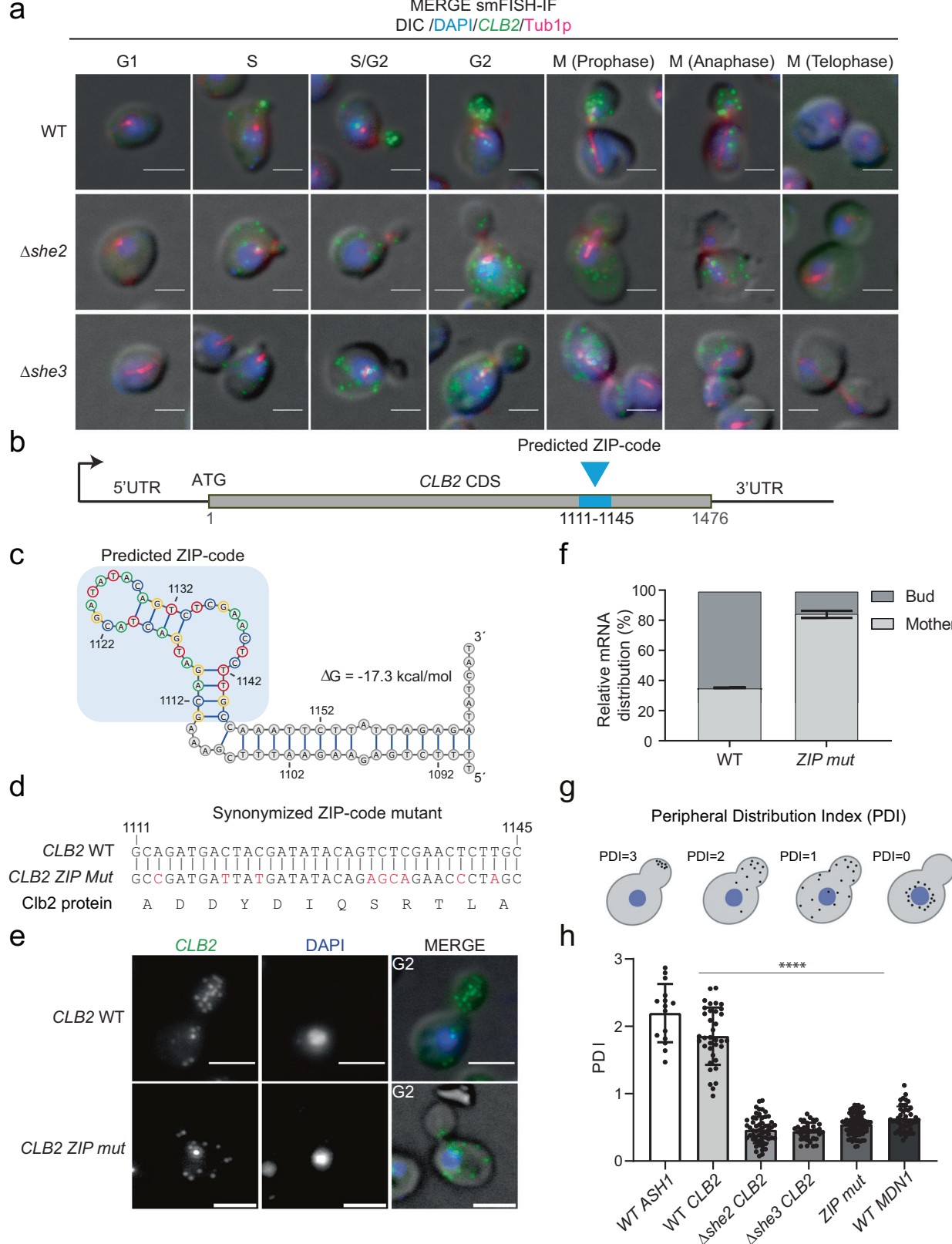

*CLB2* mRNA localization did not segregate the Clb2 protein in the daughter cell, unlike Ash1[3,7,71], but rather influenced protein levels.

## *CLB2* mRNA and protein co-localization suggest more efficient translation in the bud

To simultaneously measure *CLB2* mRNA and protein expression in single cells, we generated a strain where 25 myc tags were inserted at the N-terminus of the endogenous *CLB2* gene (Fig. 4a). This N-terminal tagging and amplification strategy increased the Clb2 protein signal without significantly affecting growth or protein stability (Supplementary Fig. 6a–d). Next, smFISH and IF were combined to simultaneously detect *CLB2* mRNAs and proteins in fixed cells, and IF against tubulin was used to score the cell cycle phase. As immunofluorescence can be noisy in yeast, we tested the specificity of the anti-myc antibody

**Fig. 2 | The She2-3 complex and an RNA ZIP code in *CLB2* mRNA CDS are required for bud localization. a** smFISH-IF in WT, *Δshe2*, and *Δshe3* strains. MERGE Maximal projections of *CLB2* mRNA smFISH (green), anti-tubulin IF (magenta) and DAPI (blue) merged to a single DIC section (gray). Cell cycle phase indicated atop the panel. Scale bars 3 μm. Representative images from at least 3 biological replicates. **b** Schematic of *CLB2* mRNA coding sequence. Blue box represents the ZIP-code at nucleotides 1111-1145 (relative to the START codon). **c** Predicted secondary structure of ZIP code (blue box) with flanking sequence (nt 1089–1168). The free energy (ΔG) of mRNA folding is indicated. **d** Synonymized ZIP-mutant (*ZIP-mut*). Top sequence: *CLB2* WT. Bottom sequence: synonymized sequence. Mutated nucleotides are indicated in red. Below is the corresponding Clb2 protein amino acid sequence (identical for both WT and synonymized strains). Rare codons were avoided to maintain the same codon usage frequency = 0.74 for the *CLB2* mRNA sequence. **e** smFISH in WT and ZIP strain. Maximal projections of smFISH with *CLB2* probes (green), DAPI (blue) and fluorescence images overlapped to a single DIC

section (MERGE). Cell cycle phase is indicated in the picture. Scale bars 3 μm. **f** Relative bud vs mother distribution of *CLB2* mRNA in WT and ZIP strain based on smFISH data like the one shown in (**e**). This data includes all cells in a budded state. Error bars indicate mean ± SD from 3 biological replicates. **g** Schematic representation of mRNA peripheral distribution index (PDI). Black dots: mRNA, blue: nucleus. A PDI close to 0 indicates that the RNA is localized near the nucleus. A PDI of 1 indicates that the RNA is diffusely dispersed throughout the cell. As the PDI value grows >1, the polarization of the mRNA increases. **h** PDI in WT cells for *ASH1*, *CLB2* and *MDN1* mRNAs and in *Δshe2*, *Δshe3* and ZIP strains for the *CLB2* mRNA. Index values are calculated from smFISH-IF experiments shown in (**e**) and Supplementary Fig. 1c, Supplementary Fig. 4a. Error bars indicate mean ± SD of cells pooled from 3 biological replicates. Significance was tested by Ordinary One-way ANOVA (For all the comparisons, $P < 0.0001$). Source data are provided as a Source Data file.

on WT cells (Supplementary Fig. 6e), which demonstrated low levels of non-specific binding of primary and secondary antibodies. This approach showed that the bulk of Clb2 proteins accumulated in the mother (M) nucleus from G2 to mitosis (Fig. 4b, c), at the time when the mRNA was preferentially found in the bud (B) (Fig. 4b, d). This indicated that Clb2 proteins were efficiently localized in the mother nucleus, as observed by live imaging (Fig. 3g) and as shown previously[29,31]. In addition, Clb2 protein foci were also found in the bud in close proximity to *CLB2* mRNAs from G2 to mitosis, suggesting that these foci may represent sites of mRNA translation (Fig. 4b, yellow arrowheads). Quantification of co-localized single mRNAs and protein foci (within 250 nm, the resolution of our system) revealed that in WT cells, more co-localized mRNA-protein foci were found in the bud than in the mother cell, while in the localization mutant *Δshe2*, mRNA-protein foci were preferentially found in the mother cell where the bulk of mRNAs was distributed (Fig. 4e, f). Furthermore, we found a significant reduction of the percentage of bud-localized mRNAs co-localized with protein foci in *Δshe2* cells compared to WT cells (Fig. 4g, non-parametric Mann-Whitney test, $P < 0.0001$), indicating that in localization mutants, *CLB2* mRNA translation efficiency may be reduced. It is interesting to note that even in WT cells, only about 25% of the bud-localized mRNAs were found in close proximity to protein foci, suggesting that *CLB2* mRNAs are translated unfrequently, as previously reported[72]. Altogether, these data suggest that in WT cells, *CLB2* mRNAs were more efficiently translated in the bud, where the mRNA was actively transported. Furthermore, in *Δshe2* cells, we observed an increase of mRNA-protein foci in the mother cell relative to the bud (Fig. 4f), despite protein levels being decreased under these conditions (Fig. 3c, d, Supplementary Fig. 6f), indicating that translation in the mother cell may be less efficient, resulting in reduced Clb2 protein levels in the localization mutants.

**Translation regulators Puf6, Ssd1, and Khd1 do not affect Clb2 protein expression**

To investigate if *CLB2* translation efficiency was higher in the bud as a result of translation repression prior to localization, we tested if the deletion of the RNA-binding proteins Puf6[12,13,17], Khd1[14–16] and Ssd1[19–21], previously shown to inhibit translation of bud-localized mRNAs, influenced Clb2 protein levels. A western blot of the endogenously myc-tagged Clb2 protein in *Δssd1, Δkhd1*, and *Δpuf6* strains (Supplementary Fig. 6g–i) did not reveal a significant increase in Clb2 protein levels when comparing the mutant strains to WT (ANOVA statistical test: $F_{(3, 8)} = 0.7677$, $p = 0.3$), which we would have expected if these proteins were inhibiting *CLB2* mRNA translation. Additionally, because Khd1 and Puf6 were shown to be important for *ASH1* mRNA localization[13,14], we tested *CLB2* mRNA localization in the mutants. smFISH for the *CLB2* mRNA in the *Δssd1, Δkhd1*, and *Δpuf6* strains showed that the mRNA was still localized in the bud at a similar level as WT cells. Altogether, these results indicated that the factors that

control *ASH1* mRNA expression were not required to control *CLB2* mRNA translation or localization. While other yet unidentified factors may exist, our data suggested the possibility that *CLB2* mRNA might also be translated outside of the bud, albeit at a reduced rate.

**CLB2 mRNA ZIP-mutant displays increased budded phase duration and cell size at birth**

The results so far showed that the absence of *CLB2* mRNA bud localization correlated with reduced protein expression, likely due to decreased mRNA translation. To corroborate this finding, we performed time-lapse microscopy. In all strains, the Clb2 protein was tagged with yeGFP while Cdc10 was fused to mCherry to label the bud neck and monitor division time and parent-offspring relationships. We confirmed nuclear accumulation of the Clb2 protein during the budded phase of single cells (Fig. 5a). Nuclear fluorescence peaked ca. 20 min before division in all strains (Fig. 5b, Supplementary Fig. 7a). Peak nuclear fluorescence intensity (Fig. 5b) as well as cumulative nuclear fluorescence over the entire budded phase (Fig. 5c, Supplementary Fig. 7b) was lower in all mutant strains investigated, but for the *Δshe* strains this reduction was not significant. Given the role of Clb2 protein in entry into and progression of mitosis[25,32–34,73] we hypothesized that mutants with lower Clb2 protein levels would show delayed mitotic progression. To allow accurate estimates of the length of the budded phase, we acquired images every minute. This led to slow growth phenotypes when green fluorescence was collected, prompting us to only collect brightfield and red fluorescence for these experiments. The mean values for the duration of the budded phase in the ZIP-mutant strain and the wild type differed by 5.27 min. Even though there was overlap in the distributions (Fig. 5d), bootstrap estimates (10000 runs, sample size 250) of the means showed almost no overlap (Fig. 5e), and the difference in means was overall highly significant ($p = 9.95 * 10^{-9}$, two-sided *t*-test based on the data in Fig. 5d). A similar but smaller increase in budded phase duration was also present in the *Δshe* strains (Fig. 5e). It seems therefore, that the duration of the budded phase is influenced by Clb2 protein concentration, which in turn is dependent on correct localization and efficient translation of the *CLB2* mRNA.

Since Clb2 protein concentration influenced budded phase duration, we hypothesized that bud size would likewise be affected. To relate Clb2 protein concentration to bud size, we collected snapshot images of growing yeast cells and correlated nuclear fluorescence of Clb2-yeGFP against bud size in all cells that were in the budded phase. We found a clear relationship between nuclear fluorescence and bud size in the wild type (Fig. 5f Pearson correlation coefficient 0.43, slope of linear regression 0.25), and a weaker one in the ZIP mutant strain (Fig. 5g Pearson correlation coefficient 0.45, slope of linear regression 0.13). Intriguingly, owing to targeted degradation of the Clb2 protein after mitosis[47,48], this correlation between nuclear fluorescence and bud size only held prior to mitosis. In post-mitotic wild-type cells, the correlation was weaker in both strains (Supplementary Fig. 7c, Pearson

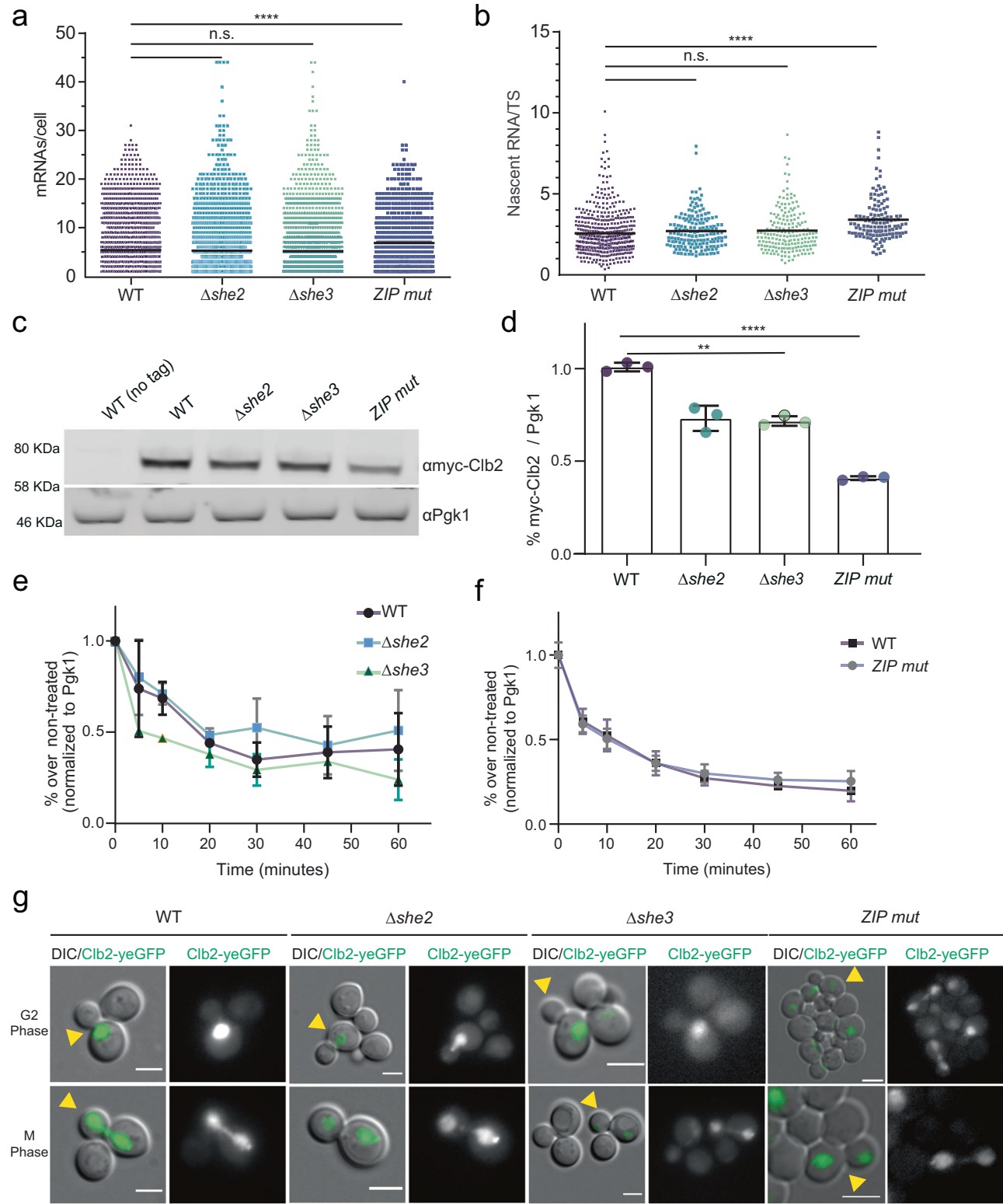

correlation coefficient 0.32 and 0.14 for WT and Zip mutant, respectively).

Given that Clb2 protein drives mitotic progression and given that Clb2 protein levels depended on intact and correctly localized *CLB2* mRNA, mutants with non-localized transcript would be expected to be born larger, as a result of the parent cell spending more time in the budded phase. Indeed, ZIP mutant cells were on average 10.9% larger at birth than wild-type cells (Fig. 5h), but no significant effect was evident for the *Δshe* mutants (Supplementary Fig. 7d, e).

## A ZIP-code rescue strain shows improved protein expression and cell cycle progression

We have shown that the ZIP code sequence located in the coding region of the *CLB2* mRNA is necessary for mRNA localization to the bud via the She2-3 complex. To test if the ZIP code needs to be in the *CLB2* coding sequence to promote efficient localization and protein synthesis, we reintroduced the minimal ZIP code sequence (36 bp) as well as an extended ZIP code region (99 bp) into the 3′UTR of the ZIP-mutant strain (Fig. 6a, Supplementary Fig. 8a). smFISH uncovered that adding

**Fig. 3 | *CLB2* mRNA mis-localization affects Clb2 protein expression but not mRNA levels, nor protein stability or localization. a** smFISH quantifications of *CLB2* mRNA expression in WT and localization mutants. Dots correspond to individual cells, from at least 3 replicates. The black bar indicates the average number of mRNAs/cell in expressing cells (WT $n$ = 3035, 5.4 ± 5.1; *Δshe2 n* = 1787, 5.3 ± 6.2; *Δshe3 n* = 1827, 5.2 ± 6.2; ZIP mutant (ZIP[1]) $n$ = 667, 6.9 ± 6.3 mRNAs/cell±SD), n.s.: not significant. Ordinary One-way ANOVA, WT vs ZIP mutant **** indicates *P*-value < 0.0001. **b** Quantification of nascent *CLB2* RNAs at transcription sites (TS) from smFISH. Dots correspond to individual cells pooled from at least 3 biological replicates (WT $n$ = 500, 2.6 ± 1.3; *Δshe2 n* = 162, 2.7 ± 1.1; *Δshe3 n* = 186; 2.7 ± 1.3, ZIP mutant (ZIP[1]) $n$ = 132, 3.4 ± 1.3 nascent RNAs/TS ± SD). Ordinary One-way ANOVA, WT vs ZIP mutant **** indicates *P*-value < 0.0001. **c** Western blot analysis using anti-myc antibody against Clb2 protein endogenously tagged with 5 myc tags in WT, *Δshe2*, *Δshe3* and ZIP mutant (ZIP[1]) cells. First lane is the control untagged strain. Endogenous Pgk1 protein was used as a loading control. **d** Quantification of the

western blot in (**c**). Myc signal normalized to Pgk1 loading control. Protein levels relative to WT are indicated. Mean ± SD from 3 replicates. Ordinary One-way ANOVA, WT vs *Δshe*2 or *Δshe*3 mutants ** indicate *P*-value < 0.01, WT vs ZIP mutant **** indicate *P*-value < 0.0001. **e** Quantifications of Clb2 protein stability assay in WT, *Δshe2* and *Δshe3* cells. Western blots (Supplementary Fig. 5f) were performed using an anti-myc antibody to detect Clb2 tagged with 5 myc tags in cells treated with 100 µg/ml cycloheximide for 0, 5, 10, 20, 30, 45 and 60 min. Pgk1 was used as a loading control. Error bars indicate mean ± SD from 3 independent experiments. **f** Quantifications of Clb2 protein stability assay in WT and ZIP mutant (ZIP[1]) as done in (**e**). Western blot (Supplementary Fig. 5g) was performed in the same way as in (**c**). Error bars indicate mean ± SD of cells pooled from 3 independent experiments. **g** Clb2 fused to yeGFP in WT, *Δshe2*, *Δshe3* and ZIP mutant cells. Maximal projections of Clb2-GFP (green) overlapped to a single DIC picture. Scale bars 2 µm. Source data are provided as a Source Data file.

both versions of the ZIP code to the 3'UTR resulted in a complete rescue of *CLB2* mRNA localization in the bud, comparable to WT levels (Fig. 6b, Supplementary Fig. 8b), demonstrating that the WT ZIP code can drive mRNA localization also from a distal location within the *CLB2* mRNA. Next, we tested whether the Clb2 protein levels were rescued in the *CLB2* ZIP rescue strain. Interestingly, only a partial rescue was observed (Fig. 6c, d, Supplementary Fig. 8c, d), suggesting that *CLB2* bud mRNA localization promoted Clb2 synthesis and contributed to controlling about 10–20% of Clb2 protein expression. Next, we tested whether the extended ZIP code would be sufficient to control bud localization and increase the expression of a non-localized mRNA. To this end, we generated a strain where the endogenous *RPT2* gene, a constitutive non-localized mRNA, was modified with the *CLB2* extended ZIP code (Supplementary Fig. 8e). As the *RPT2* coding sequence has a length (1314 bp) comparable to the *CLB2* coding sequence (1476 bp), we inserted the *CLB2* ZIP code in frame and at a similar position in the gene. Two-color smFISH performed on the *RPT2-CLB2-ZIP* strain or on the control strain using *RTP2* probes or *CLB2* probes showed that the *RPT2-CLB2-ZIP* mRNA is not bud localized (Supplementary Fig. 8f). Altogether our data showed that the short *CLB2* ZIP code is necessary but not sufficient to drive bud mRNA localization, implying that other yet uncharacterized sequences in the *CLB2* mRNA are involved in regulating mRNA localization and translation.

Finally, we investigated the growth and cell cycle distribution of the localization mutants and the ZIP code rescue strain. To measure the distribution of the three major phases of the cell cycle (G1, S, G2/M), DNA was labeled with propidium iodide and measured by flow-cytometry, followed by Gaussian mixture modeling to estimate the three cell cycle subpopulations (Fig. 6e, f, Supplementary Fig. 9a). This revealed that, while growth rates of the localization mutants were not significantly affected (Supplementary Fig. 9b), we observed a modest but significant (two-way Anova, Dunnett's multiple comparison test) increase in the G2/M fraction in the ZIP-mutant but not in the *Δshe* mutants nor the ZIP-mutant rescue strain (Fig. 6e, f). Altogether, our experiments point to a dual function of the *CLB2* mRNA ZIP code motif, promoting both efficient bud localization and translation. For the latter function to take effect, the ZIP motif needs to be in the coding sequence. Consistent with this observation, in the *Δshe* mutants where the ZIP code is still intact, higher levels of Clb2 protein were observed. In the ZIP-mutant, where both the localization is impaired and the ZIP code is absent, Clb2 protein levels were lowest in bulk populations (Fig. 3c, d) and in individuals across the cell cycle (Fig. 5b). We also observed an increased G2/M fraction in this mutant, suggesting that the duration of these stages is increased at the expense of G1 (Fig. 6e, f). Altogether, the higher G2/M fraction (Fig. 6e, f) and the larger size at budding and division (Fig. 5h) indicated that control of *CLB2* mRNA localization and translation play a role in regulating Clb2 protein levels, which in turn report translational maturity as well as size of the bud

compartment, providing a mechanism to coordinate growth and cell cycle progression during the G2/M phase transition.

## Discussion

In this study, we investigated the complete lifecycle of the *CLB2* gene by combining single-cell and single-molecule imaging methods to measure *CLB2* mRNA and protein expression throughout the yeast cell cycle.

We experimentally demonstrated that the She2-She3 complex was required to transport the *CLB2* mRNA to the bud as soon as this compartment was formed (Figs. 1, 2), and that these factors likely belong to the same pathway, as double mutant strains showed an epistatic effect (Supplementary Fig. 5c–e). Furthermore, based on mathematical modeling (Supplementary Fig. 3d–f) we proposed that active mRNA transport needs to be combined with an anchoring mechanism to explain the highly efficient (>60%) transcript localization, although anchoring factors have yet to be identified. Our observations are consistent with previous work showing that the *CLB2* mRNA can be pulled down using the She2, She3 or Myo4 proteins as bait, as well as with imaging data suggesting *CLB2* mRNA localization in the bud[22]. The She2-She3 complex localizes the *ASH1* mRNA to the bud, with the She2-mRNA interaction mediated via three *ASH1* ZIP codes (E1, E2A, E2B) located in the coding sequence and one ZIP code (E3) located in the *ASH1* 3' UTR[4–11,67]. We identified an element in the coding region of *CLB2* whose secondary structure resembles that of the *ASH1* E3 ZIP code and demonstrated that this element is necessary for transport into the bud compartment, indicating that it constitutes a bona fide She2 binding site. In *ASH1*, the ZIP code promotes localization irrespective of its position in the transcript[4,7,67,68,74,75]. We could confirm this also for *CLB2*, where moving the ZIP motif to the 3' UTR resulted in a localization pattern indistinguishable from WT (Fig. 6). Unexpectedly, insertion of the *CLB2* ZIP code into the coding sequence of the non-localized *RPT2* mRNA did not suffice to drive bud localization of the *RPT2* mRNA (Supplementary Fig. 8e, f). This finding suggested that other yet unidentified cis-acting elements within the *CLB2* mRNA are required for bud localization.

While translation of *ASH1* mRNA is tightly regulated by Puf6 and Khd1, we could find no such inhibition of translation for *CLB2* through either of these factors nor through the bud-localized translation regulator Ssd1 (Supplementary Fig. 6), although we cannot exclude regulation via yet unidentified elements. As Ash1 protein activity needs to be confined to the bud where it controls mating type switching, the reverse appears to be true for the Clb2 protein, which accumulates in the nucleus of the mother cell. Thus, while *ASH1* and *CLB2* localization appear to be controlled by the same cellular machinery, their translational regulation is different.

Alteration of the secondary structure of the ZIP motif through synonymous mutation abolished localization and led to reduced Clb2 protein levels. Moving the ZIP motif from the coding sequence to the 3'UTR rescued *CLB2* localization, but the protein concentration did not

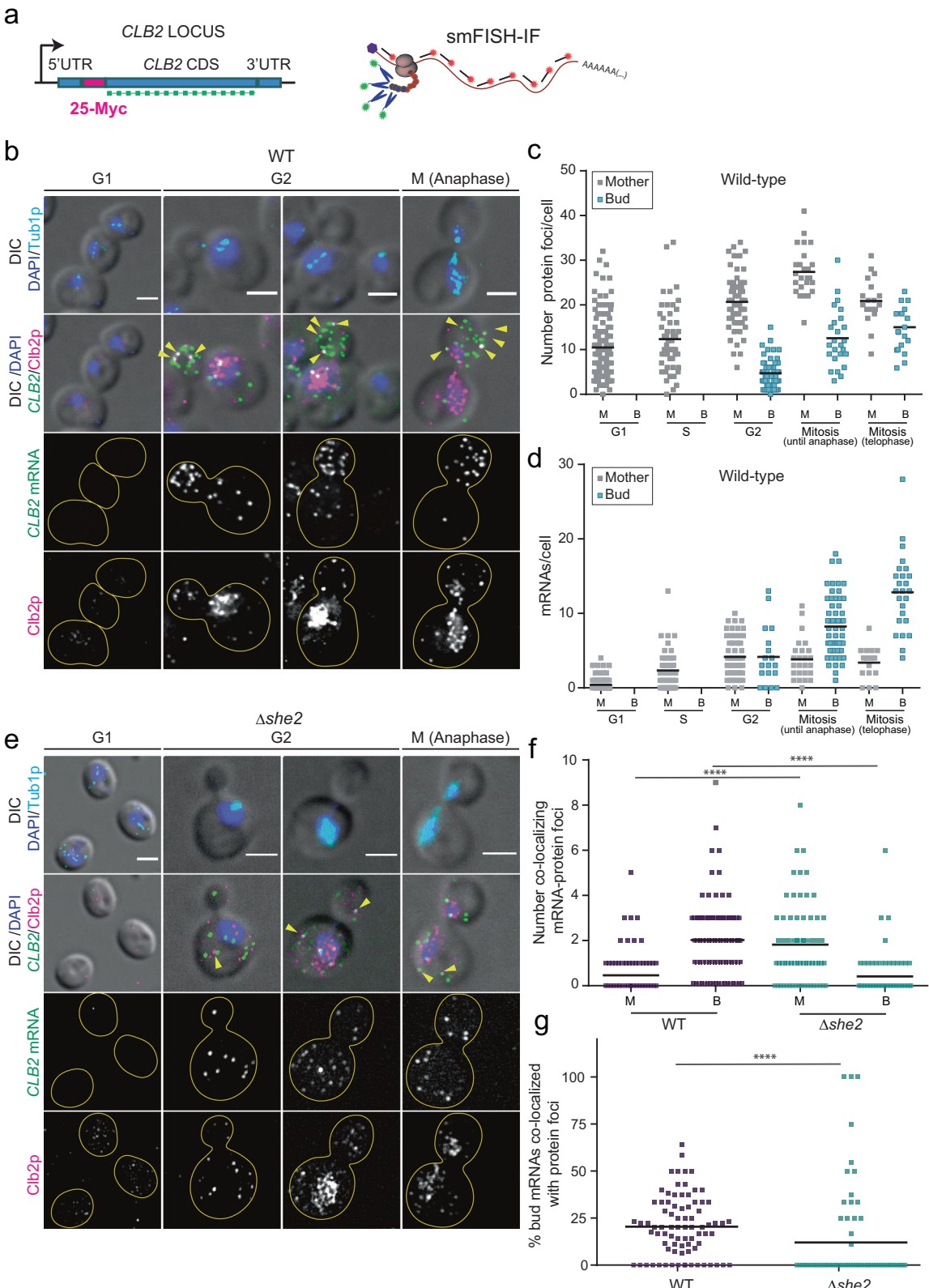

recover to WT levels. In the *Δshe* mutants, localization was abolished, but protein levels did not decrease to the levels of the ZIP mutant strain. This suggested that an intact ZIP structure located within the coding sequence of *CLB2* is required for optimal translation. We therefore propose that the *CLB2* ZIP motif fulfills a dual role—localization to the bud via She2-She3-Myo4 as well as promotion of translation through yet unidentified factors.

For the *CLB2* ZIP mutant strain, we found a 10.9% increase in the duration of the budded phase, accompanied by a 10.9% increase in birth size. While budded phase duration in the *Δshe* mutants was increased (*Δshe2*: 5.4%, *Δshe*: 6.3%), birth size was not significantly affected. The increase in bud phase duration can be explained by previous studies investigating Clb2 regulation of mitotic progression. Clb2-Cdc28 can be inactivated through phosphorylation, delaying

**Fig. 4 | *CLB2* mRNA and protein colocalization suggests more efficient translation in the bud. a** Schematic of the *CLB2* endogenous gene tagging to simultaneously visualize single mRNA by smFISH and Clb2 proteins by IF against the myc tag in fixed cells. 25 myc tags inserted at the beginning of the *CLB2* CDS, after the ATG. **b** smFISH-IF in WT cells. Top panels: MERGE Maximal projections of IF anti-tubulin (cyan) and DAPI (blue) merged to a single DIC section (gray). Second panel row from the top: MERGE Maximal projections of IF anti-myc-Clb2 protein (magenta), *CLB2* mRNA smFISH (green) and DAPI (blue) merged to a single DIC section (gray). The third and fourth panels from the top are the *CLB2* mRNA smFISH and Clb2 protein IF, respectively. Scale bars 3 μm. **c** Quantification of protein foci in WT bud (B) and mother (M) cells from IF experiments shown in (**a**). Cell cycle classification was performed using tubulin and DAPI as markers. The bars indicate the mean of cells pooled from 2 biological replicates. **d** Quantification of mRNA in WT bud (B) and mother (M) cells from smFISH experiments shown in (**a**). Cell cycle classification was performed as in (**b**). Bars indicate the mean of cells pooled from 2 biological replicates. **e** smFISH-IF in *Δshe2* cells. Maximal projections of IF anti-myc-Clb2 protein (magenta), *CLB2* mRNA smFISH (green) and DAPI (blue) merged to a single DIC section (gray). Panels description as in (**a**). Scale bars 2 μm. **f** Quantification of *CLB2* mRNA-protein foci found in proximity (<250 nm) by smFISH-IF experiments performed in bud (B) and mother (M) cells of WT and *Δshe2* cells, shown in panels (**a**, **d**). The bars indicate the mean of cells pooled from 2 biological replicates. WT vs *Δshe2* mutant **** indicate *P*-values < 0.0001 calculated using the Non-parametric T-test (Mann-Whitney). **g** Quantification of bud-localized mRNAs co-localizing with protein foci (<250 nm) in WT and *Δshe2* cells is shown in panels (**a**, **d**). The bars indicate the mean of cells pooled from 2 biological replicates. WT vs *Δshe2* mutant **** indicates *P*-value < 0.0001 calculated by using the Non-parametric T-test (Mann-Whitney). Source data are provided as a Source Data file.

mitotic progression. The phosphorylation state is set by the relative activity of the Swe1 kinase and Mih1 phosphatase[73,76]. Given that Clb2-Cdc28 can likewise inactivate Swe1 through phosphorylation, Clb2-Cdc28 could titrate against Swe1 for mitosis to progress. In mutants with lower Clb2 levels, this would happen later, increasing the duration of G2/M. Swe1/Mih1 activity ratios would set the threshold for bud compartment size, akin to a mechanism[33] recently proposed for *Schizosaccharomyces pombe*, where cell size control is mainly realized in G2.

*S. cerevisiae* mainly exerts size control in G1[63,77], however, experimental observations[78] and mathematical models[79] suggest a second size control mechanism. A mechanism in which the localization of the *CLB2* transcript reports bud compartment maturity was explored previously through mathematical models[49]. In these models, G2/M duration is more variable in the absence of mRNA localization. Our data show that nuclear Clb2 correlates with bud size (Fig. 5f, g) and that this relationship is reduced in the ZIP mutant, suggesting that nuclear Clb2 protein levels can report bud maturity. This correlation is weaker during mitosis (Supplementary Fig. 7c), which is expected as Clb2 protein is actively degraded[47,48] and is consequently less informative of bud size.

While our data points to a role of *CLB2* mRNA localization to control its translation and not Clb2 protein nuclear localization (Fig. 3), we cannot exclude that a transient localization of Clb2 in the bud may have spatially defined functions that contribute to the observed phenotypes. Previous work showed that Clb2 can also be found at the bud neck, when overexpressed[29,30] or when Clb2 nuclear import is blocked[31] and that forcing the Clb2 protein into the nucleus can lead to detrimental effects[80]. Future studies may uncover additional roles for Clb2 in the bud compartment and at the neck.

The overall small effects on budded phase duration are likely due to the fact that in the ZIP code mutant, about 40–50% of the Clb2 protein is still expressed. This might be at the lower edge of the threshold that suffices to drive mitotic entry and progression[25,32–36] which would help to put the smaller increase of the budded phase duration in the *Δshe* mutants into perspective. In these mutants, protein levels were reduced, yet there was still significant overlap with WT cell levels (Supplementary Fig. 7).

Altogether, our data indicate that coupling *CLB2* mRNA bud localization with protein synthesis allows cells to coordinate growth with cell cycle progression. We propose a model whereby active localization of the transcript to the bud compartment may lead to preferential translation of the mRNA, creating a dependency chain in which the nuclear concentration of the resulting protein reflects bud size or budding time duration. Further experimental work is needed to test these hypotheses. Furthermore, by shuttling back to the mother nucleus, Clb2 protein levels may signal when the bud is ready for mitosis, thereby establishing biochemical communication between distinct subcellular compartments.

## Methods

### Yeast cell cultures

Yeast cells were grown in the appropriate media depending on the experiment. For yeast transformation, cells were grown in YEPD (10 g/L yeast extract, 20 g/L peptone, 20 g/L glucose). Liquid cultures for smFISH, protein extractions, flow cytometry, and live imaging were done in Synthetic Complete medium (6.7 g/L yeast nitrogen base, 20 g/L glucose and synthetic complete amino acid and vitamins mix) or in Drop-out medium (6.7 g/L yeast nitrogen base, 20 g/L glucose and drop-out mixture lacking leucine or uracil). Drop-out media lacking leucine was used for the experiments shown in Fig. 1e–g, Supplementary Fig. 3b, c, to maintain the expression of the MCP-2xyeGFP plasmid (pET296). Drop-out media lacking uracil were used for transformation selection during strain construction.

Agar plates 2% (w/v) were made with YEPD or drop-out medium lacking leucine or uracil. For selection of strains expressing the hygromycin or kanamycin markers, cells were plated on YEPD plates with 300 μg/mL of hygromycin or kanamycin antibiotics. Cells were grown at the indicated temperature using constant shaking at 210 rpm. For smFISH, live imaging and flow cytometry, the details of the cell cultures are described below.

### Plasmids construction

The plasmids encoding the *CLB2* gene N-terminally tagged with myc tags were built as follows. The *CLB2* gene was cloned, including its 5′ UTR, CDS, and 3′UTR (−788 bp before the ATG, +346 bp after the STOP codon) in a YIplac211 vector. To obtain a strain where different N-terminal tags could be inserted in the coding sequence, a gene fragment was synthesized with a BamHI site after the *CLB2* ATG start codon. This fragment was sub-cloned into the YipLac211 *CLB2*-containing plasmid by using restriction enzymes SalI and BrGI. To generate the 5-myc and the 25-myc tagged *CLB2* variants, a gene fragment was synthesized encoding 5 synonymized myc tags flanked by BamHI and BglII restriction sites. The myc tags (5 or 25) were cloned into the *CLB2* gene at the BamHI site, giving rise to plasmids pET479 and pET491, respectively. The plasmids were cut with restriction enzymes AatII and BglII to generate a cassette including the modifications as well as the sequence for the homologous recombination.

The synonymized *CLB2* ZIP code plasmids were generated by Gibson cloning using the NEB Gibson Assembly Cloning kit (#E5510) using oligos (OAM32, 35, 43, 44, 70, 71, 74, 65) and plasmids (pAM90-97) indicated in Supplementary Data 1. The plasmids were cut using AatII and EcoRI-HF restriction enzymes, and the modified *CLB2* inserts were transformed into BY4741 for homologous recombination. The ZIP-code rescue plasmids (pAM097 36-nt ZIP or pET586 99-nt ZIP) were generated by inserting the predicted WT She2 binding site sequence (AGCAGATGACTACGATATACAGTCTCGAACTCTTGCC) at the ClaI restriction site 43 nucleotides downstream of the *CLB2* stop codon. The plasmids used to tag the *RPT2* gene (pET 576 and pET578)

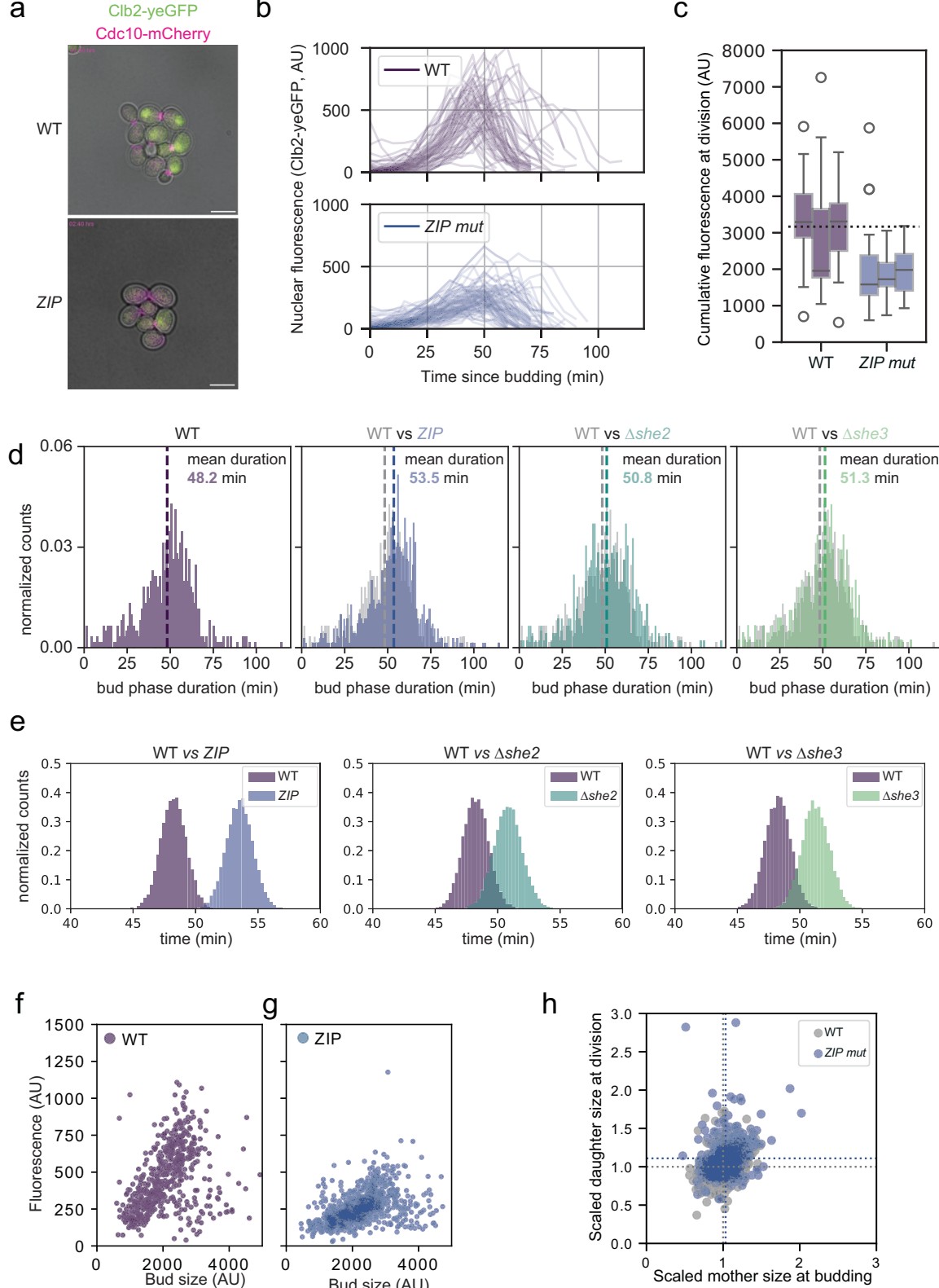

were generated as follows. A gene fragment was synthesized (Gen-Script), which included 378 nt before the STOP codon of the CDS of the *RPT2* gene, followed by 5 synonymized myc tags, a STOP codon and the 150 bp *RPT2* gene 3'UTR (pET576). Alternatively, another gene fragment was synthesized, including the same feature as pET576, plus at position 133 of the *RPT2* sequence, the 99-nt CLB2 ZIP code was inserted (pET578). Both plasmids also included the URA3 marker for

genomic insertion selection and downstream of the URA marker 70 bp of sequence homologous to the *RPT2* region downstream of the 3'UTR.

**Yeast strains construction**

All strains described in this work were derived from the *S. cerevisiae* background BY4741 (MATa; his3Δ1; leu2Δ0; met15Δ0; ura3Δ0). Strains, plasmids and primers used for strain construction are listed in

**Fig. 5 | *CLB2* mRNA ZIP code mutant displays increased duration of budded phase, loss of communication between parent and bud compartment and larger bud size. a** Snapshots from Videos 3 (WT) and 6 (ZIP-mutant, ZIP[1]) from live cell imaging of *CLB2* protein endogenously tagged with yeGFP (green) and Cdc10 tagged with mCherry (magenta). Scale bars 5 µm. **b** Average fluorescence of Clb2-yeGFP of the 10% brightest pixels per cell as a proxy for nuclear fluorescence over time. Traces shown are recorded between budding and division for cells born during the experiment and are pooled from three biological replicates. AU: arbitrary units (**c**) Boxplot showing cumulative fluorescence signal for the trajectories in (**b**), $P = 4.3 * 10^{-12}$, One-way ANOVA. Individual boxplots correspond to a single biological replicate. Horizontal line denotes the median, the box denotes the quartiles, and the error bars denote the rest of the distribution. **d** The budded phase duration as inferred from the time from appearance to disappearance of clear, neck-located Cdc10-mCherry signal. Mean durations and *p*-values (two-sided *T*-test, compared to WT): WT: 48.2 min, $n = 602$, ZIP mutant: 53.5 min, $n = 694$,

$P = 9.95 * 10^{-9}$, $\Delta she2$: 50.8 min, $n = 457$, $P = 0.013$, $\Delta she3$: 51.3 min, $n = 442$, $P = 3.59 * 10^{-3}$. Data showed is pooled from three biological replicates per strain. **e** Bootstrap results of the data in panel (**d**). From each distribution in (**d**), 250 trajectories were sampled, and the mean duration was calculated. This was done 10000 times. Nuclear fluorescence correlates with bud compartment size in WT (**f**) but not in the ZIP code mutant (**g**). Datapoints shown are restricted to pre-mitotic budded cells, that is cells that had no discernible fluorescence signal or a circular signal in the parent cell. Cells with elliptical signals or with signal in both parent and bud were excluded from this plot and are in Supplementary Fig. 7c. **h** Scaled daughter size at birth vs scaled daughter size at division shown for WT and the ZIP-mutant strain. In both strains, daughter size weakly correlates with mother size (Pearson correlation WT: 0.32 ($N = 431$), ZIP:0.25, ($N = 428$). Dotted lines indicate the median of the data, divided by the median of the WT strain (ZIP-mutant mother size at budding: 1.03, daughter size at birth: 1.109). Data shown were pooled from three biological replicates per strain. Source data are provided as a Source Data file.

Supplementary Data 1. Yeast strains were constructed with PCR amplification and transformation protocols detailed in[56]. *CLB2* was tagged with 24 × MBSV6 in the 3' UTR right after the STOP codon. The MBSV6 insert, followed by the kanamycin resistance gene flanked by LoxP sequences, was PCR amplified from plasmid pET264 with primers OET017 and OET254, each containing about 70 nucleotides of homology sequence for the *CLB2* gene. The cassette (1–2 µg) was transformed in BY4171 using a lithium-acetate protocol we previously optimized[56]. In strains, where the MS2V6 cassette was integrated into the genome, the kanamycin resistance gene was then removed by expressing the CRE recombinase under the control of the *GAL1* promoter (plasmid pET184). Obtained strains were tested by PCR with test primers (Supplementary Data 1) followed by sequencing as previously described[56]. Deletion strains for genes *SHE2*, *SHE3*, *SSD1*, *KHD1* and *PUF6* were done by PCR amplifying from plasmid a resistance marker, hygromycin or kanamycin (pET148 and pET152, respectively). The following primers, which included about 70 bp of homology sequence for the target gene, were used for the amplification: *SHE2* (OET164 and OET165), *SHE3* (OET460 and OET461), *SSD1* (OET548 and OET549), *KHD1* (OET530 and OET531) and *PUF6* (OET534 and OET535). The amplified cassettes were transformed into BY4741 and plated overnight on YEPD plates and then replica-plated on selective plates. Endogenous gene tagging of *CLB2* with GFP at the protein C-terminus was done by PCR amplifying a cassette including the GFP gene and kanamycin resistance gene from plasmid pET143 with primers OET452, OET453. Following a similar procedure, endogenous C-terminal tagging of *WHI5* with 6 HA tags was done with primers OET451 and OET289 and plasmid pET152, while tagging of *CDC10* with tdTomato was done with primers OET476 and OET477, and plasmid pET341. The RPT2 strains (YET1079, YET1082) were generated by cutting plasmids pET576 and pET578 with restriction enzymes MscI/AgeI before tranformatin of the purified cassette in BY4741 (YET181) and selection of uracil-lacking agar plates. Test primers to check the correct integration of the cassettes are listed in Supplementary Data 1.

### smFISH probes design

*CLB2* probes were designed using the Stellaris™ Probe Designer by LGC Biosearch Technologies and purchased from Biosearch Technologies. *ASH1, MDN1* and *MBSV6* probes design was previously described in[55,56]. Probes sequence and fluorophores are provided in Supplementary Data 1.

### Single molecule fluorescence in situ hybridization (smFISH)

Single-molecule FISH (smFISH) was performed as follows. Yeast strains were grown overnight at 26 °C in synthetic medium with 2% glucose and containing the appropriate amino acids to complement the strain auxotrophies. In the morning, cells were diluted to an $OD_{600}$ of 0.1 and allowed to grow until an $OD_{600}$ of 0.3–0.4. Cells were fixed by adding

paraformaldehyde (32% solution, EM grade; Electron Microscopy Science #15714) to a final concentration of 4% and gently shaken at room temperature (RT) for 45 min. Cells were then washed three times with buffer B (1.2 M sorbitol and 100 mM potassium phosphate buffer pH 7.5) and resuspended in 500 µL of spheroplast buffer (buffer B containing 20 mM VRC (Ribonucleoside–vanadyl complex NEB #S1402S), and 25 U of Lyticase enzyme (Sigma #L2524) per OD of cells (-10⁷ cells) for about 7–8 min at 30 °C. Digested cells were washed once with buffer B and resuspended in 1 mL of buffer B. 150 µL of cells were seeded on 18 mm poly-L-lysine treated coverslips and incubated at 4 °C for 30 min. Coverslips were washed once with buffer B, gently covered with ice-cold 70% ethanol and stored at −20 °C. For hybridization, coverslips were rehydrated by adding 2 × SSC at RT twice for 5 min. Coverslips were pre-hybridized with a mix containing 10% formamide (ACROS organics #205821000)/2xSSC, at RT for 30 min. For each coverslip, the probe mix (to obtain a final concentration in the hybridization mix of 125 nM) was added to 5 µL of 10 mg/mL *E. coli* tRNA/ssDNA (1:1) mix and dried with a speed-vac. The dried mix was resuspended in 25 µL of hybridization mix (10% formamide, 2 × SSC, 1 mg/ml BSA, 10 mM VRC, 5 mM NaHPO₄, pH 7.5) and heated at 95 °C for 3 min. Cells were then hybridized at 37 °C for 3 h in the dark. Upon hybridization, coverslips were washed twice with pre-hybridization mix for 30 min at 37 °C, once with 0.1% Triton X-100 in 2 × SSC for 10 min at RT, once with 1 × SSC for 10 min at RT and once with PBS 1x for 10 min at RT. Coverslips were quickly dipped in 100% ethanol and let dry at RT covered form light. Finally, coverslips were mounted on glass slides using ProLong Gold antifade (4',6-diamidino-2-phenylindole) DAPI to counterstain the nuclei (Thermofisher, #P36935).

### smFISH-IF

smFISH-IF was performed as previously described in[52,53]. In brief, smFISH-IF was performed in a similar way to smFISH, described above. After the last 1 × PBS wash of the smFISH, IF was performed on the same coverslips. The smFISH was fixed in 4% PFA in PBS for 10 min at RT and then washed for 5 min at RT with 1 × PBS. The coverslips were blocked with 1 × PBS, 0.1% RNAse-free Bovine Serum Albumin for 30 min at RT before being incubated with primary antibodies (Thermofisher, mouse anti-tubulin, 1:1000; Sigma mouse monoclonal anti-myc clone 9E10, 1:1000, Covance, mouse monoclonal anti-HA, 1:1000) in 1 × PBS, 0.1% RNAse-free Bovine Serum Albumin for 45 min. After being washed with 1 × PBS for 5 min at RT, the coverslips were incubated with the secondary antibody (goat anti-mouse Alexa 647 1:1500, or goat anti-mouse Alexa 488 1:1500) in 1 × PBS, 0.1% RNAse-free Bovine Serum Albumin for 45 min at RT. Next, the coverslips were washed with 1 × PBS three times for 5 min to remove excess antibody. Coverslips were dehydrated by dipping them into 100% ethanol and let dry before mounting onto glass slides using ProLong Gold antifade solution with DAPI.

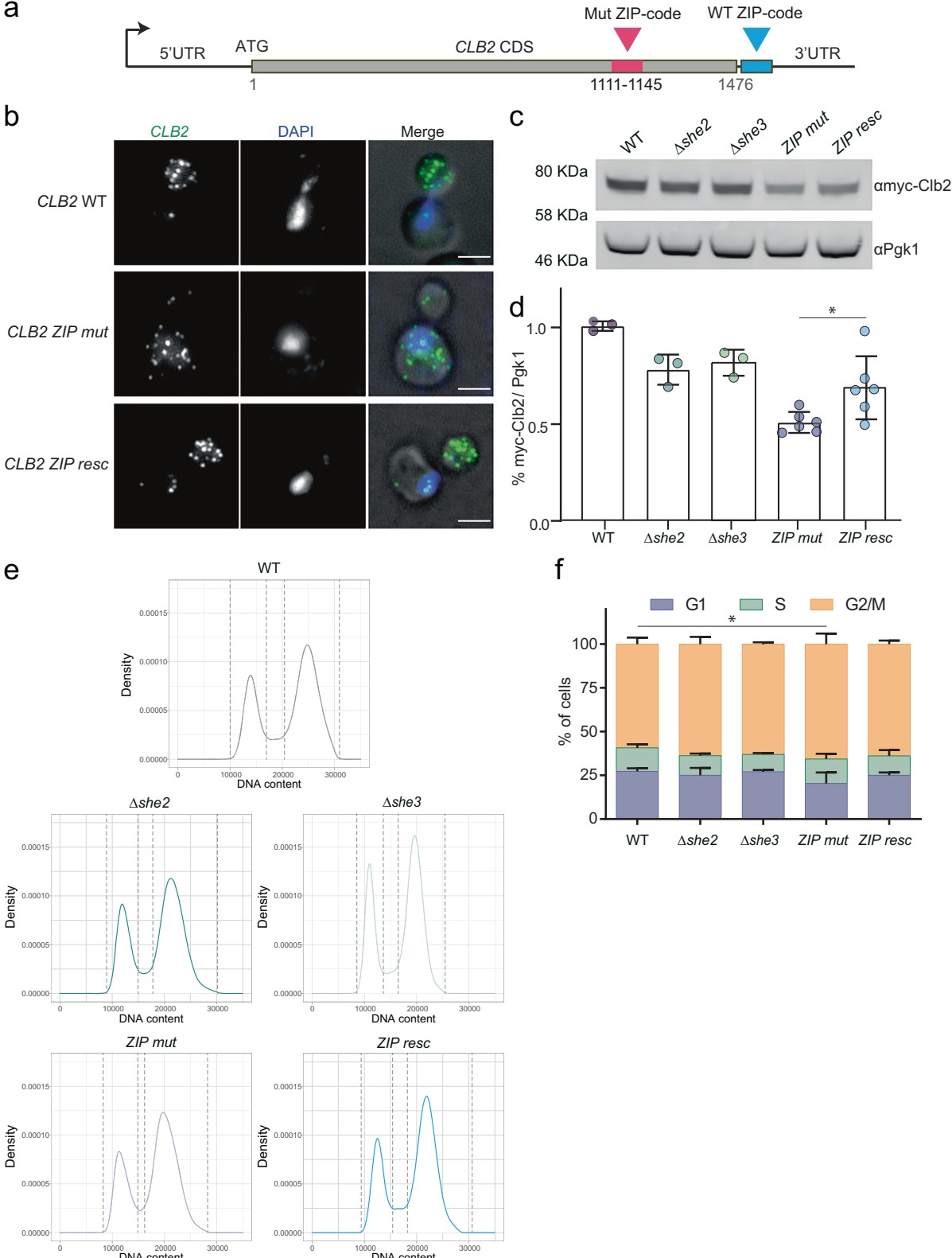

## smFISH and smFISH-IF image acquisition and analysis

Images were acquired using an Olympus BX63 wide-field epi-fluorescence microscope with a 100X/1.35NA UPlanApo objective. Samples were visualized using an X-Cite 120 PC lamp (EXFO) and the ORCA-R2 Digital CCD camera (Hamamatsu). Image pixel size: XY, 64.5 nm. Metamorph software (Molecular Devices) was used for acquisition. Z-sections were acquired at 200 nm intervals over an optical range of 8 μm. FISH images were analyzed using FISHQUANT[81]. Briefly, after background subtraction, the FISH spots in the cytoplasm were fit to a three-dimensional (3D) Gaussian to determine the coordinates of the mRNAs. The intensity and width of the 3D Gaussian were filtered to exclude nonspecific signal. The average intensity of all the mRNAs was used to determine the intensity of each transcription site.

**Fig. 6 | The *CLB2* ZIP mutant shows a small but significant cell cycle defect, and the ZIP code rescue strain partially recovers protein expression and cell cycle progression. a** Schematic of *CLB2* ZIP code rescue strain. The blue box represents the WT ZIP code inserted 41 nt after the stop codon in the 3'UTR. The Pink box represents the mutated ZIP code at nucleotides 1111–1145 (nucleotide number relative to the START codon). **b** smFISH in WT, ZIP code and ZIP code rescue strain. Maximal projections of smFISH with *CLB2* probes (green) and DAPI (blue), and fluorescence images overlapped. Scale bars 3 µm. Representative images from samples collected from at least 3 biological replicates. **c** Western blot analysis using anti-myc antibody against Clb2 protein endogenously tagged with 5 myc tags in WT, *Δshe2*, *Δshe3*, ZIP mutant and ZIP code rescue cells. First lane is the control untagged strain. Endogenous Pgk1 protein was used as a loading control.

**d** Quantification of Western blot in (**c**). Myc signal normalized to Pgk1 loading control. Protein levels relative to WT are indicated. Mean ± SD from at least 3 biological replicates. Asterisk represents significance of $P = 0.026$ as determined by Non-parametric *T*-test (Mann-Whitney). **e** Cell cycle analysis by DNA content estimation with flow cytometry in WT, *Δshe2*, *Δshe3*, ZIP mutant and ZIP code rescue cells. DNA was stained with propidium iodide to measure the G1 and G2/M components, respectively. **f** Relative distribution (%) of cells in G1, S and G2/M phase obtained by performing mixed Gaussian fitting to estimate the three subpopulations. Mean ± SD from 3 experiments. Statistical significance was calculated using Two-way ANOVA. Two-ways ANOVA, WT vs ZIP mutant * indicates *P*-value $P < 0.05$. Source data are provided as a Source Data file.

## Quantification of peripheral distribution index

The peripheral distribution index (PDI) was quantified as described in[70]. Briefly, the Matlab-based software RDI (RNA dispersion index) calculator was used to calculate the peripheral distribution index for each cell by identifying cellular RNAs and describing their distribution in relation to the nucleus. Prior to analysis with the RDI calculator, the RNA channel was processed using a 3D Laplacian of Gaussian filter of radius = 5 and standard deviation = 1. The cell and nucleus channels were processed using the brightness/contrast function in ImageJ to enhance the contrast between the object and the background, as advised in[70].

## Co-localization analysis

For the co-localization of *CLB2* mRNA and protein foci in Fig. 4, the FISH-quant data for the individual molecules were used as x, y, z coordinates, and Euclidean distances for all protein-mRNA molecule combinations were calculated in the mother and daughter cells. Protein and mRNA molecules closer than 250 nm were considered to be in a translation complex. Multiple protein molecules can be within 250 nm of a single mRNA molecule, and this would still be considered a single translation complex.

## PDE solution for mRNA diffusion

We use a modified diffusion equation at steady state to model the mRNA movement in terms of diffusion of a concentration $c(x,y,z)$ in 3 spatial dimensions, and include binding to ribosomes (uniformly spread) leading to the formation of complexes $b(x,y,z)$:

$$0 = D \cdot \nabla^2 \mathbf{c}(x,y,z) - k_d \cdot \mathbf{c}(x,y,z) + k_p - k_{on} \cdot \mathbf{c}(x,y,z) + k_{off} \cdot \mathbf{b}(x,y,z) \tag{1}$$

$$0 = k_{on} \cdot \mathbf{c}(x,y,z) - k_{off} \cdot \mathbf{b}(x,y,z) - k_d \cdot \mathbf{b}(x,y,z) \tag{2}$$

With decay constant $k_d = \mathrm{Ln}(2)/t_{0.5} = \mathrm{Ln}(2)/240 \ \mathrm{s}^{-1}$, $k_{on} = 0.25 \ (k_{off} + k_d) = 0.0035$ and $k_{off} = 1/90 = 0.011$ chosen to reflect a half-life of 240 s, a 90 s mean lifetime of ribosome-bound complexes, and that approximately 20% of mRNAs appear bound at steady state. In the high-binding scenario, $k_{on}$ was increased by a factor of 125. For the numerical implementation, the production constant is represented by a small non-zero spread around the bud center using a smooth step-function of which the volume integral is normalized to 1:

$$k_p = \frac{k_{max} \cdot e^{s \cdot m}}{e^{m \cdot r} + e^{s \cdot m}} \tag{3}$$

For the simulation results shown in Supplementary Fig. 3d–f, the values chosen are $s = 5$ and $m = 10$, and the normalization results in $k_{max} = 0.0019$. Although the exact value of this constant affects the absolute concentration of mRNA, it does not affect the ratio of mother to bud RNA. The PDE is solved in three Cartesian coordinates using the

FEM implementation in Wolfram Mathematica (Wolfram Research, Inc., Mathematica, Version 12.3.1, Champaign, IL).

## Ellipsoid fitting to mother and bud DIC images

Differential interference contrast (DIC) images were analyzed in Mathematica to fit 3D ellipsoids to mother and bud models. X- and y-axes lengths were measured for the cells, and the short axis was used as an estimation of the z-axis. The z-axis origin value was aligned with the z-stack images by maximizing the FISH-Quant mRNA and protein point inclusions.

## Pattern search to predict ZIP codes and synonymization

To identify potential ZIP codes in the *CLB2* mRNA, we performed a targeted pattern search[69]. In the first step, we leniently screened for two nested pairs of inverted repeats with a minimal length of four nucleotides that framed an asymmetric bulge region as found in the E3 ZIP code in the *ASH1* mRNA[66]. We also checked for the presence of a CGA motif and a singular cytosine on the opposite strand with a defined distance of six nucleotides[75]. The search was performed on the complete *CLB2* mRNA with 1476 nt of CDS (YPR119W; genomic coordinates: chromosome XVI, 771653-773128, +, genome version S288C; *Saccharomyces* Genome Database, https://www.yeastgenome.org/locus/S000006323). We further added 366 nt 3' UTR and 346 nt 5' UTR as previously determined by[46]. In the second step, the minimum free energy (MFE) folds of all initial instances were analyzed using RNAfold with and without including a constraint on the nested pairs of inverted repeats[82,83]. Fold prediction was performed at 28 °C with 80 nt RNA sequence fragments centered on each instance. Instances were only kept if (i) at least one of the inverted repeat pairs was present in the MFE structure without constraints, and (ii) the free energy (ΔG) of the constraint structure did not differ by more than 20% from the MFE structure without constraints. The latter accounts for energetic benefits from interaction with the She2-She3 proteins. The pattern search predicted a single ZIP code at nucleotide positions +1111 to +1145 of the CDS (genomic coordinates: chromosome XVI, 772763-772797, +). Figure 2c displays the predicted fold with constraint using RNAfold of the ZIP code instance plus ±5 nt flanking sequence. Visualization of the predicted structure in dot-bracket notation was generated using VARNA[84]. Repeating the pattern search described above on the synonymized ZIP code mutant did not retrieve any hits. The complete sequences of the synonymized ZIP code mutant are provided in Supplementary Data 1.

## Sample preparation for live yeast fluorescence imaging

Yeast cells were grown at 26 °C in synthetic selective medium. Exponentially growing cells (OD$_{600}$ 0.2–0.4) were plated on coated Delta-T dishes (Bioptech 04200417 C). The dishes coating was done by incubating with Concanavalin A 1 mg/ml (Cayman Chemical Company) for 10 min at RT. Excess liquid was aspirated, and dishes were dried at RT. To activate Concanavalin A, dishes were incubated for 10 min at RT with a 50 mM CaCl$_2$, 50 mM MnCl$_2$ solution. Excess was removed and dishes dried at RT. Finally, dishes were washed once with ultrapure

water (Invitrogen) and completely dried at RT. Cell attachment was performed by gravity for 20 min at RT, excess liquid removed and substitution with fresh media. Cells were diluted to an $OD_{600}$ of 0.1 and grown until $OD_{600}$ 0.3–0.4. before being plated on a Concanavalin A-coated dish.

## Single molecule mRNA live cell fluorescence imaging and image analysis

The two-color simultaneous imaging of mRNAs and the appropriate cellular marker was performed on a modified version of the home-built microscope described in[55,56]. Briefly, the microscope was built around an IX71 stand (Olympus). For excitation, a 491 nm laser (CalypsoTM, Cobolt) and a 561 nm laser (JiveTM, Cobolt) were combined and controlled by an acoustic-optic tunable filter (AOTF, AOTFnC-400.650-TN, AA Opto-electronic) before being coupled into a single-mode optical fiber (Qioptiq). The output of the fiber was collimated and delivered through the back port of the microscope and reflected into an Olympus 150 × 1.45 N.A. Oil immersion objective lens with a dichroic mirror (zt405/488/561rpc, 2 mm substrate, Chroma). The tube lens (180 mm focal length) was removed from the microscope and placed outside of the right port. A triple band notch emission filter (zet405/488/561 m) was used to filter the scattered laser light. A dichroic mirror (T560LPXR, 3 mm substrate, Chroma) was used to split the fluorescence onto two precisely aligned EMCCDs (Andor iXon3, Model DU897) mounted on alignment stages (x, y, z, θ- and φ- angle). Emission filters FF03-525/50-25 and FF01-607/70-25 (Semrock) were placed in front of green and red channel cameras, respectively. The two cameras were triggered for exposure with a TTL pulse generated on a DAQ board (Measurement Computing). The microscope was equipped with a piezo stage (ASI) for fast z-stack and a Delta-T incubation system (Bioptech) for live-cell imaging. The microscope (AOTF, DAQ, Stage and Cameras) was automated with the software Metamorph (Molecular Devices). For two-color live-cell imaging, yeast cells were streamed at 50 ms, the Z plane was streamed, and z-stacks were acquired every 0.5 μm. Single-molecule analysis was done on maximal projected images using Fiji. Maximally projected images were filtered using the Maxican Hat filter (Radius = 2) in Fiji. Spots were identified and counted using the spot detection plugin integrated in TrackMate. LoG detector was used for the spot identification, object diameter = 3 and Quality threshold = 2500. Files were exported as csv files and plotted using GraphPad Prism.

## Deconvolution algorithm

To reduce imaging artifacts arising from noise and optics of the microscope, we used the Huygens software v3.6, where a Classic Maximum Likelihood Estimation (CMLE) algorithm was applied as a restoration method to deconvolve the images used for protein-mRNA foci co-localization (Fig. 4). CMLE assumes the photon noise to be governed by Poisson statistics and optimizes the likelihood of an estimate of an object in the input 3D image while taking the point spread function into consideration. The CMLE deconvolution method was chosen since it is suited for images with low signal-to-noise ratio and for restoring point-like objects. The result is a more accurate identification of the location of the object, which in our case is the fluorescently labeled mRNA and protein molecules. The restoration parameters used with the CMLE deconvolution algorithm were 99 iterations, a quality stop criterion of 0.01, and a signal-to-noise ratio of 15.

## *CLB2* mRNA bud localization quantification in living cells

For the analysis reported in Supplementary Fig. 3c, the ImageJ plugin Labkit (https://imagej.net/Labkit) was manually used to segment cells and RNAs. Segmented cells were used as input for training the deep learning program Stardist in 2 dimensions. Stardist was used to automatically detect and segment cells and single mRNAs from live imaging movie frames (Cell Detection with Star-convex Polygons, https://arxiv.org/pdf/1806.03535.pdf). Cell and RNA segmentation was imported into R using the RImageJROI package. In R, the cell size, number of mRNAs in the bud and the distance of each bud-localized mRNA to the periphery was calculated and plotted over time using the R packages Spatial Data and PBSmapping[85]. The Stardist segmentations were used to plot the RNAs and the cell's periphery onto the live imaging movie using the FFmpeg wrapper function for the FFmpeg multimedia framework (https://ffmpeg.org/).

## Protein extraction and Western blot

Yeast strains were grown overnight at 26 °C in yeast extract peptone dextrose (YEPD) medium with 2% glucose. In the morning, cells were diluted to an $OD_{600}$ of 0.1 and allowed to grow until $OD_{600}$ 0.5–1. Cell lysis was performed by adding 1 ml $H_2O$ with 150 μL of Yex-lysis buffer (1.85 M NaOH, 7.5% 2-mercaptoethanol) to the pellet of 3–5 ODs of cells (~$3 \times 10^7$) and kept 10 min on ice. Proteins were precipitated by the addition of 150 μL of TCA 50% for 10 min on ice. Cells were pelleted and resuspended in 100 μL of 1X sample buffer (1 M Tris-HCl, pH 6.8, 8 M Urea, 20% SDS, 0.5 M EDTA, 1% 2-mercaptoethanol, 0.05% bromophenol blue). Total protein extracts were fractioned on SDS-PAGE and examined by Western blot with mouse anti-myc (Sigma), mouse anti-Pgk1 (Thermofisher). For quantitative Western blot analyses, fluorescent secondary α-Mouse (IRDye 800CW) and α-Rabbit (IRDye 680RD) antibodies were used. The signals were revealed using the LYCOR® scanner and quantified using LITE® Software.

## Clb2 protein imaging in live cells

Cells were precultured in liquid SC medium containing 2% glucose (w/v). Exponentially growing cells were transferred to SC medium containing 2% glucose (w/v), solidified with 1.5% low-melting agarose in an ibidi u-Slide 4-well sample holder. Cells were imaged at 30 °C using a Nikon Ti-eclipse widefield fluorescence microscope equipped with a SOLA 6-LCR-SB light source (Lumencor), an Andor Zyla 5.5 sCMOS camera, and a Plan Apo lambda 100x immersion Oil objective (NA 1.45, refractive index 1.515). Brightfield images were collected with 20 ms exposure at an LED (CooLED) intensity of 7.6. Clb2-yeGFP was imaged with a 480/40 excitation filter, a 535/50 emission filter and a 505 nm longpass dichroic (Semrock). Cdc10-mCherry was imaged with a 570/20 excitation filter, a 610 nm longpass emission filter, and a 600 nm longpass dichroic (Semrock). Because of the sensitivity of the yeast strains to the GFP imaging conditions, images were acquired with different LED strengths and exposure, depending on the experiment. For the snapshot experiments (nuclear fluorescence vs bud size), Clb2-yeGFP was imaged at 100% SOLA intensity with 750 ms exposure and no binning, Cdc10-mCherry was imaged at 20% SOLA intensity with 500 ms exposure and no binning. For 5 min time-lapse experiments, Clb2-yeGFP was imaged at 10% SOLA intensity with 200 ms exposure and 4 × 4 hardware binning. Cdc10-mCherry was imaged at 10% SOLA intensity with 100 ms exposure and 4 × 4 hardware binning. For 1 min timelapse experiments, Clb2-yeGFP was not imaged. Cdc10-mCherry was imaged at 10% SOLA intensity with 100 ms exposure and 4 × 4 hardware binning.

## Image processing

Bright-field images were segmented using a Cellpose model, and the association of objects in subsequent frames (tracking) was achieved with a maximum matching approach. Briefly, a graph was constructed with cell detections as nodes. Edges were drawn between nodes in subsequent frames if the corresponding cells were less than a pre-defined threshold (200 px) apart. Edges were given weights computed as 1/(node distance), and then the edge set was found that connected each node to exactly one node in the subsequent frame and that minimized the total distance. Bud necks were detected with a unet-

based multiclass detector, which was trained to recognize background, cytosol, neck outline and neck centers. The training data for the detector was generated through semi-automatic annotation of Cdc10-mCherry fluorescence images using a napari-based custom interactive random forest plugin. Neck centers in different frames pertaining to the same mother-bud pair were associated into tracks using the tracking approach described above. Lengths of the neck tracks were reported as the budded phase duration. Budding frames were determined as the first time-point at which a fluorescent Cdc10-RFP patch could be detected. In some cases, the emergence of a bud as judged from the bright-field image preceded the formation of a detectable fluorescent patch. In these cases, the budding frame was adjusted manually based on the bright-field information. Pixel areas of mother cells in the budding frame were reported as the size of the mother compartment at budding. Pixel areas of mother and daughter cells in the first frame after a bona fide neck could be detected were reported as birth sizes of the respective compartments. For the snapshot experiments, parent/offspring associations were assigned as follows: Bud neck quality was predicted from red fluorescence images (Cdc10p-mCherry) using a random forest classifier, and matches were reduced to matches that contained bona fide necks. Nuclear accumulation of Clb2 was judged from the green and the red channels with a random forest classifier. In ambiguous cases (more than two objects overlapping with a neck), a Bayesian approach was followed to disambiguate: Self-matches were given probability zero, as a cell cannot be its own parent or child. All other matches were considered equiprobable a priori. When an object had a fluorescent nucleus, it was considered twice as likely to be a parent.

Object size was incorporated as follows:

$$P(\text{object1} = \text{parent}, \text{object2} = \text{bud}|\text{area1} = a_1, \text{area2} = a_2) \propto \frac{a_1}{a_1 + a_2} \quad (4)$$

Overlaps were incorporated as follows:

$$P(\text{object1} = \text{parent}, \text{object2} = \text{bud}|\text{overlap1} = o_1, \text{overlap2} = o_2)$$
$$= P(\text{object1} = \text{bud}, \text{object2} = \text{parent}|\text{overlap1} = o_1, \text{overlap2} = o_2) \propto \frac{o_1 + o_2}{\sum_k o_k} \quad (5)$$

Following this, the parent-bud pair with maximum posterior probability was selected. The following Python libraries were used during data processing: hydra[86], cellpose[87], pandas[88], numpy[89], matplotlib[90], napari[91], astropy[92], scikit-learn[93], scikit-image[94], photutils[92].

## Growth curves setup and analysis
Cells were grown overnight at 30 °C in SC with 2% glucose. Cells in mid-log phase were spun down, the supernatant was removed, and cells were resuspended at a final $OD_{600}$ of about 0.1 in SC with 2% glucose. In 48-well plates with flat bottom, 400 µL were plated per well. At least 3 well replicates were done per experiment. Cells were grown for the indicated time at 30 °C. $OD_{600}$ measurements were taken every 5 min with 700 rpm orbital shaking between time-points using a CLARIOstar® plate reader (BMG Labtech). Growth curves analysis was performed using an adaptation of the R package Growthcurver[95] and plotted using the R package ggplot2[96], tydiverse[97], RColorBrewer[98], dplyr[99]. Growthcurver fits a basic form of the logistic equation to experimental growth curve data. The logistic equation gives the number of cells $N_t$ at time t.

$$N_t = \frac{K}{1 + \left(\frac{K - N_0}{N_0}\right)e^{-rt}} \quad (6)$$

The population size at the beginning of the growth curve is given by $N_0$. The maximum possible population size in a particular environment, or the carrying capacity, is given by $K$. The intrinsic growth rate of the population, $r$, is the growth rate that would occur if there were no restrictions imposed on total population size. The best values of $K$, $r$, and $N_0$ for the growth curve data were found using the implementation of the non-linear least-squares Levenberg-Marquardt algorithm. The carrying capacity and growth rate values ($K$ and $r$) were used to compare the growth dynamics of the strains.

## Flowcytometry sample preparation and analysis
Cells were grown overnight at 30 °C in SC medium with 2% glucose. In the morning, cells were diluted to an $OD_{600}$ of 0.1 and were grown to mid-log phase ($OD_{600}$ 0.3–0.4) with constant shaking (200 rpm) at 30 °C. When cells had reached mid-log phase, 1 mL of culture was transferred to a 1.5 mL Eppendorf tube and centrifuged for 3 min at 845 g. The supernatant was removed, and cells were resuspended in 70% ethanol and fixed overnight at 4 °C. Cells were washed once with 1 × PBS pH 7.4, resuspended in 500 µL of 1 × PBS with 1 µL of RNAse A (1 mg/mL) and incubated at 37 °C for 2 h. After incubation, cells were washed with 1 mL of 1 × PBS and resuspended in 200 µL of 1 × PBS. The resuspended cells were divided into two tubes: 94 µl of cells were supplemented with 6 µL of a 1 mg/mL solution of propidium iodide (PI), obtaining a final concentration of 60 µg/mL PI, while another 100 µL of cells remained unstained as a negative control. Both negative control and PI-stained cells were incubated in a water bath at 30 °C for 1 h, covered from the light. Cells were then washed 3 times with 1 mL 1 × PBS and resuspended in 500 µl of 1 x PBS. The cells were analyzed with the Beckman Colter CytoFLEX S Flow Cytometer (B2-R0-V2-Y2). A 561 nm laser was used to excite the PI-stained cells, and a band-pass filter (610/20 nm) was used to filter the emitted fluorescence. 50'000 cells were collected per sample. Analysis and plotting was performed using R Studio and the following R packages: ggplot2[96]; tydiverse[97], RColorBrewer[98], dplyr[99], mixtools[100].

## Statistics and reproducibility
For all the smFISH images reported in the manuscript, we selected representative images from at least 2 biological replicates. All images compared side-by-side in the same panel were processed in the same way (contrast and brightness adjustments done in Fiji). FISH-quant was used to quantify single mRNA molecules and protein foci in fixed samples. For mRNA and protein measurements, the analysis was performed on the raw, unfiltered images, then processed automatically and in batch using FISH-quant. No statistical method was chosen to determine the sample size of fixed cell data. For all the experiments, the Investigators were not blinded to allocation during experiments and outcome assessment. Fiji was used to quantify single mRNA molecules in living cells. GraphPad Prism 10 was used to calculate the mean and the standard deviation (SD) of all the data and perform statistical analysis. Flowcytometry data, growth curves analysis was performed in R Studio, as detailed in previous paragraphs. For each experiment, the number of biological replicates, the number of cells analyzed (n), the statistical analysis applied, and significance are indicated in the figure, figure legend or in the main text. Symbols meaning: Not significant (n.s.) $P > 0.05$; $P < 0.05$; *; $P < 0.01$, **; $P < 0.001$, ***; $P < 0.0001$, ****. No statistical method was chosen to determine the sample size of live-cell data. Cells/colonies that were out of focus were excluded from analysis because they could not be segmented in a satisfactory fashion. Investigators were blind to which strain was analyzed during manual annotation of budding/division events. All other processing was automated, ensuring unbiased quantification.

## Reporting summary
Further information on research design is available in the Nature Portfolio Reporting Summary linked to this article.

## Data availability

The raw data plotted in this manuscript can be found in the Source Data file. In addition, information about the reagents (i.e., plasmids, yeast strains, primers, smFISH probes) used in this study can be found in Supplementary Data 1 associated with this manuscript. Both the Source Data file and Supplementary Data 1 are also saved in the Zenodo repository https://doi.org/10.5281/zenodo.16032630 [https://zenodo.org/records/16032630]. Due to size (several TB), raw microscopy data are available from the corresponding author upon request. Source data are provided with this paper.

## Code availability

The code used to process flowcytometry, growth and Clb2 protein expression measurements by time-lapse microscopy can be found in the Zenodo repository https://doi.org/10.5281/zenodo.16032630 [https://zenodo.org/records/16032630].

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

## Acknowledgments

This work was supported by NIH grant GM57071 to R.H.S. E.T. was supported by Swiss National Science Foundation Fellowships P2GEP3_155692 and P300PA_164717, as well as by the Vrije Universiteit Amsterdam. We acknowledge financial support from the Department of Science and Technology/National Research Foundation in South Africa (grant NRF-SARCHI-82813 to J.L.S.) and grant 116298 (to D.D.v.N.). K.Z. was supported by the Deutsche Forschungsgemeinschaft (DFG) via the Research Unit FOR2333 (ZA 881/3-1). We thank X. Meng for help with cloning. J. Wiemerink for help with flow cytometry. M. Collart (CMU, Geneva) for the *RPT2* smFISH probes. A. Gerber for discussion and critical reading of the manuscript, W. Li, C. Eliscovich, F. Bruggeman, S. Das and all the members of the lab for discussion.

## Author contributions

Conceptualization, E.T.; Methodology, E.T., A.M., P.S., J.L.S., D.D.N., and K.Z.; Formal Analysis, E.T., A.M., J.L.S., M.S., D.D.N., and P.S.; Investigation, E.T., A.M., P.S., and K.R.; Writing – Original Draft, A.M. and E.T.; Writing –Review & Editing, E.T., A.M., R.H.S., P.S., K.Z., and J.L.S.; Funding Acquisition, E.T., J.L.S., K.Z., and R.H.S.; Supervision, E.T. and R.H.S.

## Competing interests

The material in this manuscript (MS2 system V6) is the subject of a patent application to the US Patent and Trademark Office (no. 18242217). It has not been licensed to any corporation, and the authors (R.H.S., E.T.) are the inventors together with Maria Vera Ugalde (McGill University). The remaining authors declare no competing interests.
