## [Transparent Peer Review file · Nature Communications]

Cyclin *CLB2* mRNA localization and protein synthesis link cell cycle progression to bud growth

Corresponding Author: Dr Evelina Tutucci

Version 1:

Reviewer comments:

Reviewer #1

(Remarks to the Author)

In this manuscript, Maekiniemi and colleagues report a novel role for CLB2 mRNA trafficking and localized translation in coordinating cell cycle progression and bud growth in budding yeast. The authors used single-molecule RNA FISH and live-cell imaging to show that the CLB2 mRNA accumulates at the bud of yeast cells in a cell-cycle dependent manner. While Clb2 protein is synthesized in the bud, it does not accumulate in the daughter nucleus. The authors identified a zipcode element within the coding sequence of the CLB2 mRNA and show that mutations that disrupt the She2/She3 binding motif in the zipcode inhibit CLB2 mRNA localization and decrease Clb2 protein expression. Interestingly, deletion of either She2 or She3, or moving the zipcode to the 3' UTR of a zip-mutated CLB2 mRNA, only partially rescue optimal expression of the Clb2 protein, suggesting that the zipcode has yet another undefined function in regulating Clb2 protein synthesis. Finally, expression of a zipcode-mutated CLB2 results in larger daughter cells and loss of synchronization between growth and cell cycle.

Although a role for CLB2 mRNA bud-localization as a sizer of bud growth has already been suggested from mathematical modeling (Spiesser et al., PLOS Comp. Biol, 2015), this manuscript provides the first experimental evidence supporting a mechanism that coordinates growth and cell cycle progression during G2/M phase transition. As such, this work is an important contribution to the field. The results provided are of high quality, especially those from the various single-cell and single-molecule imaging approaches, and their quantitative image analyses.

Furthermore, the authors provide convincing evidence that the CLB2 zipcode fulfills two roles: promote mRNA localization via the She2-She3-Myo4 pathway; and promote CLB2 expression. However, the most important role of the zipcode seems to be linked to its regulation of CLB2 expression, since the she2/she3 mutants have a minor effect on cell size. While the mechanism of CLB2 mRNA localization has been well characterized in the manuscript, how this zipcode promotes Clb2 expression remains unknown. The authors argue that the zipcode must be in the coding sequence to promote optimal translation, but another possibility is that the zipcode is part of a larger cis-acting element that promotes mRNA translation. This would explain why insertion of the zipcode alone in the 3' UTR of the CLB2 ZIP mutant mRNA or deletion of She2/3 only partially rescue protein expression (Figure 6c-d). Since such dual role for a zipcode is unusual, it would have been nice to have a better characterization of the mechanism of zipcode-mediated regulation of Clb2 expression (i.e polysome gradient, identification of separation of function mutations,...).

Major comments:

- Figure 4C: It is surprising that there's so many Clb2 protein foci in G1 mother cells knowing that this protein is actively degraded in G1. Since there's a destruction box element at the N-terminus of Clb2, is it possible that the N-terminal 25 myc-tags may affect the stability of this protein? The authors should look at the degradation of the 25myc-Clb2 protein, as it may be disrupted.

- This raises the question why the authors inserted their myc tags at the 5' end of CLB2 ORF, which is not commonly used in other publications on Clb2. Since the effect of the she2/3 mutants or zipcode mutation of Clb2 expression is small (25% to 50% reduction), a validation of Figure 3D using an anti-Clb2 antibody to detect the untagged protein would help strengthen their conclusions.

- The authors should validate that the CLB2 zipcode is sufficient to control the localization and translation of a reporter mRNA, independently of the context of the CLB2 mRNA.

- Figure 6D: Is it possible to rescue Clb2 protein expression using the zipcode from another bud-localized transcript, like ASH1 for instance? What happens if the ASH1 E3 zipcode is inserted in the 3'UTR of the CLB2 ZIP mut mRNA? Is it a unique role for the CLB2 zipcode to promote protein expression?

Minor comments:

There are a few mistakes in the text, such as "allows" instead of "arrows" (page 5, line 16).

Page 15, line 9: "The She2-She3 complex localizes the ASH1 mRNA to the bud, with the She2-mRNA interaction mediated via the E3 ASH1 ZIP code located in the ASH1 coding region" ASH1 has 4 zipcodes, and not only one. The E3 zipcode is in the 3'UTR.

Page 15, line 17: "Puf1" should be replaced by "Puf6".

Page 53, line 9, Legend video 2: "One single Z-stack was streamed every 50 ms". This is not clear. Is that mean a single plane was captured every 50 ms or a Z-stack was captured every 50ms? Also, at page 7, line 4, the Video 2 is described in the main text as "High frame-rate imaging every 100 ms..." Please clarify.

(Remarks on code availability)

Reviewer #2

(Remarks to the Author)

Localization of mRNAs can regulate translation and protein localization. The authors here confirmed that CLB2 mRNA localizes to the bud of yeast cells and seek to understand how this may regulate Clb2 protein levels and localization. They find that CLB2 mRNA localization is dependent on She2 and She3, like the well-studied, bud-localized ASH1 mRNA. The authors identify a ZIP code in the coding sequence of CLB2 mRNA that is necessary for mRNA bud localization. RNA binding proteins that inhibit ASH1 mRNA translation were not found to be important for CLB2 mRNA regulation, overall demonstrating key similarities and differences in the mRNA level regulation of CLB2 and ASH1.

To address the function of CLB2 mRNA localization to the bud, the authors suggest that 1) CLB2 mRNA translation is more efficient in the bud and 2) CLB2 mRNA localization to the bud contributes to cell size regulation. While these ideas are very compelling, more evidence is needed to support these conclusions.

Major comments

1. To provide evidence for CLB2 mRNA translation being more efficient in the bud, the authors demonstrate a slight but significant reduction in Clb2 protein expression observed via Western Blot in *she2Δ* and *she3Δ* cells and a further reduction in a ZIP code mutant. However, localization to the bud is not sufficient for efficient CLB2 translation since bud localization was rescued with a ZIP code in the 3'UTR but protein levels hardly increased. The authors suggest that the ZIP code in the coding region may be important for efficient translation through some other mechanism, but, alternatively, it could be that localization to the bud does not regulate CLB2 translation very much at all. To further test whether the ZIP code has an effect on translation efficiency separate from localization, the authors could put the ZIP code mutant (with and without 3'UTR Zip rescue) in *she2Δ* cells. The prediction would be that Clb2 protein levels are further reduced than in cells with *she2Δ* alone.

2. The authors provide a secondary line of evidence that bud-localized CLB2 mRNA is more efficiently translated in the bud based on the co-localization of CLB2 mRNA and Clb2 protein foci based on immunofluorescence staining (IF). IF in budding yeast is notoriously tricky, and indeed the pattern of Clb2 protein in live cell images (Fig 3g) is very different from the IF images (Fig 4b and 4e). In particular, the foci that are used for measurements of co-localization with mRNA in Fig 4 are not present in Fig 3. To strengthen this data, the authors need to show that the Clb2 protein foci in the cytoplasm are not artifacts of IF.

3. Based on the extensive literature of Clb2-CDK driving mitotic entry and progression, there is a clear prediction that lower Clb2 protein will slow G2/M progression resulting in larger cells at the time of cell division. This is supported by the data in Fig 5 where the ZIP code mutants with lower Clb2 protein levels show larger cells at division. However, this does not explain why mother cells would be larger at the time of budding (Fig 5C), which is controlled by the G1 size control mechanism and at a time when no Clb2 is present. This also doesn't explain why more bud volume is added in G2/M without an increase in budded timing. Is it possible these cells are larger at all cell stages for other reasons?

4. The authors try to link CLB2 mRNA localization and translation efficiency to cell cycle progression and a potential bud size mechanism. The crucial measurement is described in the text as "While nuclear fluorescence tightly predicted bud size in WT and the Δ she mutants' (Fig. 5k and Extended Data Fig. 7m) this effect was significantly more dispersed in the ZIP-mutant strain, indicating a loss of communication between the two compartments." A scatter plot of nuclear fluorescence vs bud size for the two populations would be more appropriate here. The authors also note that the ZIP code mutant added

more material during the budded phase than WT (Fig 5g) and spent longer median time in budded phase (Fig 5h, albeit the authors recognize this result is non-significant). The authors conclude that “control of CLB2 mRNA translation in the bud compartment reports bud size”. These two statements (and the overall description of Fig 5) are difficult to parse, and therefore the intended conclusions and interpretation from the data in Fig 5 are unclear. The results need to be rewritten for clarity, and further experiments are needed that more directly test associations between CLB2 mRNA localization and bud growth. If the intended conclusion is that Clb2 protein levels are regulated by bud size, it would be much more convincing to include other experiments manipulating bud size (latrunculin, or mutants in polarity, vesicle delivery, or septins) and determining the effect on Clb2 protein levels.

Minor comments

5. In Fig 2 e, f, explicitly state which stage of the cell cycle is being imaged.

6. The authors assume the ZIP code mutant abrogates She2 binding, but it would be much stronger to show this directly by co-IP.

7. A more careful description of strain construction is needed. Are the 5-myc Clb2 WT and ZIP code mutants constructed at the endogenous locus and in the same way?

8. According to the morphogenesis checkpoint model, the presence of a bud is read out by a change in septin organization, which then leads to inactivation of Swe1, the Clb2-CDK inhibitor (Howell and Lew, 2012). A discussion of how the proposed regulation of Clb2 translation relates to the regulation of Clb2 protein activity by the morphogenesis checkpoint would be very useful for readers to synthesize the conclusions here.

9. In an effort to make scientific language more inclusive, the authors could consider changing references of mother and daughter cells to parent (or originator) and progeny cells (White, et al, 2021, doi: 10.1038/d41586-021-03490-7).

(Remarks on code availability)

I was able to access the raw data and code in the zenodo repository, but there were no README files that I could find. I'm not very familiar with zenodo, so maybe I just missed it.

Reviewer #3

(Remarks to the Author)

This manuscript uses a combination of fixed cell and live cell imaging to explore how localization of the CLB2 mRNA affects protein expression, protein localization, cell size, and cell cycle progression. Overall, the experiments are well done, well described, and the conclusions are supported by the data presented. Much of the paper is a beautiful example of how rigorous quantification can provide new insights into biology. The paper doesn't have all the answers and, in some respects, raises more questions than it answers. It reveals that there must be multiple mechanisms that regulate where translation takes place because translation regulation for CLB2 is very different than ASH1, another well-studied mRNA. The work is exciting in identifying a zip code in the mRNA that is important for localization and perhaps translation.

One thing I am trying to understand is: the localization mutants only show a subtle decrease in protein expression. Why is localization of the mRNA to the bud important? The authors suggest that the mRNA may be more efficiently translated in the bud than in the mother based on data in Figure 4 that show there is greater colocalization of mRNA and protein in the bud than the mother. But the difference is quite subtle (a mean of less than 1 colocalization spot in the mother compared to ~ 2 colocalization spots in the bud). While the difference is statistically significant it seems extremely small from a biological perspective. Could the authors comment on whether this observation?

In Figure 4, the authors show that protein expression increases in G2/M and protein is more frequently found in the bud (but it is still found in the mother more frequently). mRNA expression increases significantly in the bud in G2/M. In WT cells only 25% of mRNA associated with protein suggesting that mRNA is infrequently translated. In the mRNA localization mutant, the authors found more colocalizing mRNA and protein in the mother and less in bud - was this done as a function of the cell cycle (like in WT)? Did authors quantify mRNA and protein (like in c and d) and do they see a decrease in amount of protein at the single cell level (like in WB)?

The data presented in Extended Data Figure 7 are confounding. Up until this point, the data in the manuscript seemed to support the argument that the zip code in the mRNA interacts with the she2-she3 system to localize to the bud and localization to the bud affects protein translation and hence protein expression levels (to some extent). But the extended data in figure 7 suggest that delta-she mutants that prevent localization of the mRNA to the bud don't exhibit a cell growth defect. In the discussion, the authors rationalize this observation with the explanation that perhaps the zip code serves a dual function – promoting bud localization and efficient translation. If that is the case, wouldn't we expect to see differences between the delta-she2 mutant and the zip code mutant in the single cell single molecule experiments in Figure 4? As far as I can tell, these experiments weren't done with the zip code mutant. But doing so, could potentially provide experimental evidence for the authors' claims.

(Remarks on code availability)

The zenodo link provides all raw data for the paper. It also provides code (in R and Python) for data analysis of FACS data

and imaging data. As far as I can tell scripts are provided but there are no readme files. I didn't install and run the code.

Reviewer #4

(Remarks to the Author)

(Remarks on code availability)

Version 2:

Reviewer comments:

Reviewer #1

(Remarks to the Author)

The authors have properly answered the comments of the reviewers. Overall, this is a very interesting work which uses leading-edge quantitative imaging to unveil a new mechanism linking mRNA localized translation to cell growth and cell cycle progression. I think this manuscript is now acceptable for publication in this journal.

(Remarks on code availability)

Reviewer #2

(Remarks to the Author)

In this revision, the authors have greatly improved the experiments and clarity of the text. I am convinced that CLB2 mRNA localization in the bud supports efficient translation of Clb2 protein, and, consistent with the field, the authors have provided convincing evidence that Clb2 protein levels affect cell division timing. However, I remain concerned with the interpretations and conclusions surrounding CLB2 mRNA localization to the bud reflecting bud growth. This paper is strong and doesn't require any further experiments, but the interpretations in the results and discussion sections need to be more cautious.

The data is very convincing that Clb2 protein levels correlate with duration of the budded phase and cell division timing. However, the relationship between Clb2 protein levels and bud size are less straightforward. On several occasions, the authors have attempted to explain that Clb2 protein levels are a function of bud size (a couple examples below), an interpretation that is not well supported with the data in this manuscript. An alternative explanation could be that bud size is only a function of time, i.e. more time in budded phase results in bigger buds. Since lower Clb2 protein levels extend the duration of the budded phase, the bud sizes could simply be larger due to lower Clb2 protein levels. While both interpretations are possible, the data presented in the paper does not directly address this issue. As stated in the original review, if the intended conclusion is that Clb2 protein levels are regulated by bud size, it would be much more convincing to include other experiments that manipulate bud size with independent approaches (latrunculin, or mutants in polarity, vesicle delivery, or septins) and determine the effect on Clb2 protein levels. These experiments are probably outside the scope of the current manuscript, so I recommend just that the causal relationship between Clb2 mRNA localization, protein levels, and bud size be restricted to speculation in the Discussion section.

Example statements to reconsider:

- 297 "Taken together, this chain of dependences can function as a mechanism that measures maturity of the bud compartment; active localization of a transcript to the bud compartment may provide preferential access of that transcript to the ribosomes that are localized in that compartment, and a larger bud compartment harbors more ribosomes. The concentration of the resulting protein should then reflect bud size." – This argument is unsupported and speculative.
- 310 "If CLB2 mRNA localization were part of a mechanism that measures bud maturity, mutants with non-localized transcript would be expected to be born larger, as a result of the parent cell spending more time in the budded phase. Indeed, ZIP mutant cells were on average 10.9% larger at birth than wild type cells (Fig. 5h), but no significant effect was evident for the Δ she mutants (Extended data Fig. 7d,e)." – While the data supports a correlation between CLB2 mRNA localization and cell size, the text here implies causation where CLB2 mRNA measures bud maturity.
- 430 "Altogether, our data suggest that by coupling CLB2 mRNA bud localization and protein synthesis, cells coordinate cell growth with cell cycle progression by sensing the bud translation capacity via CLB2 mRNA transport and localized translation." – This point about sensing bud translation capacity is highly speculative and not a direct conclusion of the data presented.

(Remarks on code availability)

I was able to access the code and the README file.

Reviewer #3

(Remarks to the Author)

I appreciate the authors' due diligence in preparing this revised manuscript. They have satisfactorily addressed all my concerns - either through experiments or explanations. I am supportive of publication.

(Remarks on code availability)

The zenodo link provides all data and code for the paper. It now includes a readme file.

Reviewer #4

(Remarks to the Author)

(Remarks on code availability)

Reviewers' Comments and Point-by-point rebuttal

We would like to thank all the reviewers for the constructive criticism. Following the suggestions, we have significantly revised the manuscript and addressed all the referees' comments in the point-by-point rebuttal in the pages below. A summary of the key modifications is provided here:

For the revised manuscript we generated 5 new strains:

- We generated ZIP mutant strains combined with the deletions of *SHE2* or *SHE3* (**New Extended data Figure 5c,d,e**). The strains showed an epistatic effect which suggested that the CLB2 ZIP code and the SHE2/3 complex belong to the same pathway.
- We generated a CLB2 ZIP code rescue strain with an extended ZIP code region of 99 nt to test whether regions adjacent to the ZIP code loop (36 nt) also contributes to mRNA localization and translation (**New Extended data Figure 8a,b,c,d**). This revealed that the extended region also is capable of rescuing CLB2 bud mRNA localization and induce a small increase in Clb2 protein expression, comparable to the short (36 nt) ZIP code.
- We generated strains where the *CLB2* ZIP code was introduced in the coding sequence of a non-bud localized mRNA, *RPT2* (**New Extended data Figure 8e,f**). smFISH revealed that this mRNA is not bud localized, thus demonstrating that the *CLB2* ZIP code is necessary but not sufficient to drive bud mRNA localization.

For the revised manuscript we added the following controls:

- We tested a commercially available anti-Clb2 antibody to test if we could use it instead of the anti-myc antibody (**Rebuttal, page 5**). This unfortunately did not work.
- We tested whether N-terminally tagging the Clb2 protein with 25 myc tags significantly changed the protein stability compared to the 5 myc-tagged strain (**New Extended data Figure 6c,d**). Our results show no significant change in stability.
- We added a control smFISH-immunofluorescence where we verified the specificity of the anti-myc antibody (**New Extended data Figure 6e**).

For the revised manuscript we added the following new experiment:

- To more precisely quantify the duration of the budded phase in the WT and in the localization mutants, we re-shot timelapse imaging experiments with higher temporal resolution (**New Figure 5d,e,f,g**). This data, combined with an improved bud cell segmentation pipeline, revealed a significant increase in the budded phase duration both for the ZIP code and the SHE deletion mutants.

Finally, we modified the text to improve clarity and to adapt the manuscript to the journal's format requirements. All the text modifications are highlighted in yellow in the re-submitted document.

- We wrote a 150 words abstract.
- We changed the titles of paragraphs to make them shorter.
- We changed the text describing Figure 5, to improve clarity.
- We extended the discussion, as per recommendation.
- We generated a new Zenodo repository version (new doi: 10.5281/zenodo.16032630) with the updated figures raw data, code and a readme file.

Point-by-point rebuttal

Reviewer #1 (Remarks to the Author):

In this manuscript, Maekiniemi and colleagues report a novel role for CLB2 mRNA trafficking and localized translation in coordinating cell cycle progression and bud growth in budding yeast. The authors used single-molecule RNA FISH and live-cell imaging to show that the CLB2 mRNA accumulates at the bud of yeast cells in a cell-cycle dependent manner. While Clb2 protein is synthesized in the bud, it does not accumulate in the daughter nucleus. The authors identified a zipcode element within the coding sequence of the CLB2 mRNA and show that mutations that disrupt the She2/She3 binding motif in the zipcode inhibit CLB2 mRNA localization and decrease Clb2 protein expression. Interestingly, deletion of either She2 or She3, or moving the zipcode to the 3'UTR of a zip-mutated CLB2 mRNA, only partially rescue optimal expression of the Clb2 protein, suggesting that the zipcode has yet another undefined function in regulating Clb2 protein synthesis. Finally, expression of a zipcode-mutated CLB2 results in larger daughter cells and loss of synchronization between growth and cell cycle.

Although a role for CLB2 mRNA bud-localization as a sizer of bud growth has already been suggested from mathematical modeling (Spiesser et al., PLOS Comp. Biol, 2015), this manuscript provides the first experimental evidence supporting a mechanism that coordinates growth and cell cycle progression during G2/M phase transition. As such, this work is an important contribution to the field. The results provided are of high quality, especially those from the various single-cell and single-molecule imaging approaches, and their quantitative image analyses.

Furthermore, the authors provide convincing evidence that the CLB2 zipcode fulfills two roles: promote mRNA localization via the She2-She3-Myo4 pathway; and promote CLB2 expression. However, the most important role of the zipcode seems to be linked to its regulation of CLB2 expression, since the she2/she3 mutants have a minor effect on cell size. While the mechanism of CLB2 mRNA localization has been well characterized in the manuscript, how this zipcode promotes Clb2 expression remains unknown. The authors argue that the zipcode must be in the coding sequence to promote optimal translation, but another possibility is that the zipcode is part of a larger cis-acting element that promotes mRNA translation. This would explain why insertion of the zipcode alone in the 3'UTR of the CLB2 ZIP mutant mRNA or deletion of She2/3 only partially rescue protein expression (Figure 6c-d). Since such dual role for a zipcode is unusual, it would have been nice to have a better characterization of the mechanism of zipcode-mediated regulation of Clb2 expression (i.e polysome gradient, identification of separation of function mutations,..).

We would like to thank the reviewer for the positive assessment of our work and for the constructive criticism. As detailed in the point-by-point rebuttal, for the revised manuscript we generated new strains and performed experiments to further characterize the ZIP code-mediated regulation of *CLB2* expression.

Major comments:

- Figure 4C: It is surprising that there's so many Clb2 protein foci in G1 mother cells knowing that this protein is actively degraded in G1. Since there's a destruction box element at the N-terminus of Clb2, is it possible that the N-terminal 25 myc-tags may affect the stability of this protein? The authors should look at the degradation of the 25myc-Clb2 protein, as it may be disrupted.

To address the reviewer's question, we first tested whether we could use a commercially available anti-Clb2 antibody to compare the stability of the myc-tagged Clb2 proteins to the endogenous Clb2 protein. As previously commercially available antibodies are no longer available (e.g. Santacruz (sc-9071) used for example, in PMID: 34787675), we tested the antibody sold by Cusabio (<https://www.cusabio.com/Custom-Antibodies/CG22-Antibody-12888599.html#a04>).

Unfortunately, this antibody exhibited high levels of non-specific binding to unknown endogenous proteins (see image below). Protein extracts were collected from the following strains: wild-type (WT, no tag), 5-myc Clb2, 25-myc Clb2 and a strain where the *CLB2* coding sequence was replaced by another coding sequence (effectively a $\Delta clb2$ strain). The extracts were loaded in duplicate on an acrylamide gel and separated by size. After transfer on a nitrocellulose membrane, the membrane was cut along the dashed line (see picture below). On the left side (panel a, i), it is shown the membrane hybridized with the anti-Clb2 antibody, while on the right-side (a, ii), it is shown the membrane hybridized with the anti-myc antibody. On the left side, the black asterisk (*) indicates the expected size of the untagged Clb2 protein, the blue (*) indicates the expected size of the 5-myc Clb2 protein, while the orange (*) indicates the expected size of the 25-myc protein. Panel b shows the loading control, which is a comassie staining of the acrylamide gel after transfer.

a

b

Loading control - Coomassie staining

Thus, we concluded that the Cusabio Clb2 antibody was not suitable to quantify Clb2 protein expression, while the anti-myc provided a clean and specific signal.

As we could not use the anti-Clb2 antibody, we compared the stability of the 25-myc Clb2 tagged protein to the 5-myc Clb2 version. We performed protein stability assays to compare

the stability of the 5-myc Clb2 and 25-myc Clb2 proteins as we previously did for Figure 3 e,f). Cells were grown to mid-log phase and incubated with 100 $\mu\text{g}/\text{mL}$ cycloheximide to block mRNA translation. Time-points were collected at the indicated times. As shown below, the stability of the 5-myc Clb2 and 25-myc Clb2 proteins is comparable. This new figure is now included as **Extended data Figure 6, c and d**.

Altogether, the new data suggests that N-terminally tagging the Clb2 protein with 25-myc tags does not significantly affect Clb2 protein expression compared to 5-myc tags. This is consistent with the growth assays results reported in **Extended Figure 6,b**, showing no significant growth defect in the myc-tagged strains compared to each other and to the control strain, suggesting that the tagged strains can be used for cell physiology experiments. Furthermore, the observed Clb2 decay pattern is comparable the one independently reported in other publications using either tagged or untagged versions of Clb2 (Hendrickson et al. *Curr. Bio* 2001 PMID 11719221, or Bäumer et al. *Febs Letters* 2000 PMID 10692575, Schwab et al. *Cell* 1997 PMID 9288748).

Nevertheless, our control experiments do not exclude a change of stability compared to the non-tagged Clb2 protein. It is possible that myc-tagging causes an increase in Clb2 stability that could explain some of the Clb2 protein foci observed in G1 cells by FISH-IF, yet without causing a fitness decrease.

- This raises the question why the authors inserted their myc tags at the 5'end of CLB2 ORF, which is not commonly used in other publications on Clb2. Since the effect of the she2/3 mutants or zipcode mutation of Clb2 expression is small (25% to 50% reduction), a validation of Figure 3D using an anti-Clb2 antibody to detect the untagged protein would help strengthen their conclusions.

The choice of adding the myc tags at the N-terminus of the Clb2 protein was motivated by the goal of simultaneously measuring newly synthesized Clb2 proteins and the encoding

CLB2 mRNA by smFISH-IF. By adding the tags at the N-terminus, we expected to improve the visualization of co-localized mRNA and protein foci. This approach was inspired by the translation reporter systems developed in the Singer lab for mammalian cells (Wu et al. Science 2016 PMID 27313041). This system combines nascent protein imaging by N-terminally tagging a protein of interest with 24 copies of the Sun-Tag (Tanenbaum et al. Cell 2014 PMID 25307933, Yan et al. Cell 2016 PMID 27153498) and mRNA visualization using the MS2 system. While this reporter was mainly used for translation measurement in living cells, it was also used for mRNA-protein colocalization in fixed cells by smFISH-IF (Wu et al. Figure 5 and Supplementary Figure 6, PMID 27313041).

As detailed in the previous point, the use of a commercially available antibody targeting the endogenous Clb2 protein proved to be unsuccessful. Nonetheless, our data shows that the tagging the Clb2 protein with 5-myc tags provides a reliable (across tens of biological replicates) and background-free approach for measuring Clb2 protein expression, suggesting that even small Clb2 changes can be detected using an anti-myc antibody.

Importantly, we independently confirmed the Clb2 protein expression reduction in the Δ *SHE* and ZIP code mutants by using *CLB2*-GFP tagged strains and single-cell fluorescence measurements (**Figure 5b,c and Extended data figure 7a,b**). In these strains, the GFP tag is introduced at the C-terminus of the Clb2 protein, and the extent of protein reduction is consistent with the western-blot measurements.

- The authors should validate that the *CLB2* zipcode is sufficient to control the localization and translation of a reporter mRNA, independently of the context of the *CLB2* mRNA.

To address this point, we generated new strains where we first tested the ability of an extended version of the ZIP code (99 nt), to rescue *CLB2* mRNA localization and translation (**New Extended data Figure 8a,b,c,d**). This showed similar results as the minimal 36 nt ZIP code. Next, the endogenous *RPT2* gene was modified with the *CLB2* Zip code (**New Extended Figure 8e,f**). The *RPT2* gene was selected because it has a length comparable to the *CLB2* gene, and we knew from our previously unpublished smFISH results that the *RPT2* mRNA is not bud localized in *S. cerevisiae*. Thus, we inserted in the *RPT2* coding region, at a position similar to the *CLB2* ZIP code position, a *CLB2* ZIP-code variant of 99 bp. We used a longer ZIP code variant to test the importance of the *CLB2* ZIP code neighboring sequence in controlling bud mRNA localization and possibly translation (the characterization of this longer variant is reported in the New **Extended data Figure 8e,f**). The results reported below show that the 99 bp Zip code variant is not sufficient to localize the *RPT2* mRNA to the bud. Thus, it is possible that, like the *ASH1* mRNA, the *CLB2* mRNA harbors additional localization elements in the coding sequence or in the UTRs contributing to bud mRNA localization.

These interesting new results were added to the revised manuscript in **Extended Figure 8** and will be further investigated in future work.

- Figure 6D: Is it possible to rescue Clb2 protein expression using the zipcode from another bud-localized transcript, like *ASH1* for instance? What happens if the *ASH1* E3 zipcode is inserted in the 3'UTR of the *CLB2* ZIP mut mRNA? Is it a unique role for the *CLB2* zipcode to promote protein expression?

While we find that the reviewer suggests an interesting experiment, we think that this request focusses on characterizing *ASH1* ZIP code, thus it exceeds the goal of characterizing the *CLB2* ZIP code properties and function. Furthermore, our data in Extended Figure 6 f-h suggests that the translation regulation of the *ASH1* and *CLB2* mRNAs is different, thus making difficult to interpret a potential rescue of *CLB2* mRNA translation with the *ASH1* E3 ZIP code. In future studies we will continue investigating the translation regulation of bud-localized mRNAs.

Minor comments:

There are a few mistakes in the text, such as “allows” instead of “arrows” (page 5, line 16).

Thank you for spotting this typo. It is now corrected in the revised version of the manuscript.

Page 15, line 9: “The She2-She3 complex localizes the ASH1 mRNA to the bud, with the She2-mRNA interaction mediated via the E3 ASH1 ZIP code located in the ASH1 coding region” ASH1 has 4 zipcodes, and not only one. The E3 zipcode is in the 3’UTR.

Good point. This sentence was misleading. We corrected it in the revised manuscript.

Page 15, line 17: “Puf1” should be replaced by “Puf6”.

Thank you for spotting this typo. It is corrected in the revised manuscript.

Page 53, line 9, Legend video 2: “One single Z-stack was streamed every 50 ms”. This is not clear. Is that mean a single plane was captured every 50 ms or a Z-stack was captured every 50ms? Also, at page 7, line 4, the Video 2 is described in the main text as “High frame-rate imaging every 100 ms...” Please clarify.

Good point. This description was not clear. We corrected it in the revised manuscript. Indeed, we meant a single plane was captured every 50 ms.

Reviewer #2 (Remarks to the Author):

Localization of mRNAs can regulate translation and protein localization. The authors here confirmed that CLB2 mRNA localizes to the bud of yeast cells and seek to understand how this may regulate Clb2 protein levels and localization. They find that CLB2 mRNA localization is dependent on She2 and She3, like the well-studied, bud-localized ASH1 mRNA. The authors identify a ZIP code in the coding sequence of CLB2 mRNA that is necessary for mRNA bud localization. RNA binding proteins that inhibit ASH1 mRNA translation were not found to be important for CLB2 mRNA regulation, overall demonstrating key similarities and differences in the mRNA level regulation of CLB2 and ASH1.

To address the function of CLB2 mRNA localization to the bud, the authors suggest that 1) CLB2 mRNA translation is more efficient in the bud and 2) CLB2 mRNA localization to the bud contributes to cell size regulation. While these ideas are very compelling, more evidence is needed to support these conclusions.

We would like to thank this reviewer for taking the time to review our work and to provide constructive criticism. To address the reviewer's concerns, we generated new strains and performed experiments to further characterize the ZIP code-mediated regulation of *CLB2* mRNA localization and protein expression.

Major comments

1. To provide evidence for CLB2 mRNA translation being more efficient in the bud, the authors demonstrate a slight but significant reduction in Clb2 protein expression observed via Western Blot in *she2Δ* and *she3Δ* cells and a further reduction in a ZIP code mutant. However, localization to the bud is not sufficient for efficient CLB2 translation since bud localization was rescued with a ZIP code in the 3'UTR but protein levels hardly increased. The authors suggest that the ZIP code in the coding region may be important for efficient translation through some other mechanism, but, alternatively, it could be that localization to the bud does not regulate CLB2 translation very much at all. To further test whether the ZIP code has an effect on translation efficiency separate from localization, the authors could put the ZIP code mutant (with and without 3'UTR Zip rescue) in *she2Δ* cells. The prediction would be that Clb2 protein levels are further reduced than in cells with *she2Δ* alone.

The reviewer raises an important point that is central to the message of the paper. What is the contribution of mRNA localization to Clb2 protein expression? Are the ZIP code functions in translation and localization separated?

As detailed in the paper, the ZIP code rescue strain (**Figure 6**) shows a small (15-20%) but significant increase in Clb2 protein expression. We suggest that this 15-20% increase corresponds to the contribution of mRNA localization in controlling Clb2 protein expression. We suggest that the mRNA localization-dependent mechanisms of Clb2 protein expression regulation works alongside other well-known mechanisms of Clb2 protein expression regulation (including, proteasome-dependent PMID: 8020094, APC-dependent PMID: 12152084, Hct1-dependent PMID: 9288748). This point was rephrased in the discussion to improve clarity.

To address whether the ZIP code functions in translation and localization are separated we generated two new strains in which we combined the deletion of *SHE2* or *SHE3* with the ZIP code mutant. As hypothesized by the reviewer, we would expect in the double mutants a further reduction of Clb2 protein levels if the effect of the She proteins and the ZIP code on Clb2 protein expression were independent. However, our data showed an epistatic effect as in the double mutant strains (*ZIP mut + Δshe2* and *ZIP mut + Δshe3*) we did not observe a further decrease in Clb2 protein expression (Figure below and **New Extended data figure 5 c,d,e**).

Overall, the new data suggests that the She proteins act directly on the *CLB2* mRNA, even though future studies should confirm and characterize this interaction in *in vitro* and *in vivo*.

2. The authors provide a secondary line of evidence that bud-localized *CLB2* mRNA is more efficiently translated in the bud based on the co-localization of *CLB2* mRNA and Clb2 protein foci based on immunofluorescence staining (IF). IF in budding yeast is notoriously tricky, and indeed the pattern of Clb2 protein in live cell images (Fig 3g) is very different from the IF images (Fig 4b and 4e). In particular, the foci that are used for measurements of co-localization with mRNA in Fig 4 are not present in Fig 3. To strengthen this data, the authors need to show that the Clb2 protein foci in the cytoplasm are not artifacts of IF.

The reviewer raises a good point, which gives us the opportunity to discuss the Clb2 protein expression results in more detail.

Firstly, we only partly agree with the statement that “the pattern of Clb2 protein in live cell images (Fig 3g) is very different from the IF images (Fig 4b and 4e)”. In both experimental setups we see that the Clb2 protein is mainly nuclear localized, suggesting that both tagging methods (N-terminal and C-terminal) preserve the expected Clb2 nuclear localization. However, we agree that the immunofluorescence data in fixed cells looks spottier than the GFP data collected in living cell. We think that this spotty profile is due to the primary and secondary antibodies signal amplification, which is even further amplified by the presence of 25-myc tags in the Clb2 protein. This could potentially allow the detection of single Clb2 proteins.

To address the reviewer’s concern, we added the requested control in the new **Extended data Figure 6e**. We did not add it earlier to keep the manuscript concise, but we agree that this is an important technical control that should be in the paper. As shown below, we performed immunofluorescence using the anti-myc antibody both in wild-type cells and the

25-Clb2 myc-tagged strain. The experiments shown below were done on the same day and the data were collected using the same imaging conditions. Furthermore, to compare the two images side-by-side, the same contrast and brightness levels were used. The immunofluorescence shows low levels of non-specific binding in wild-type cells, suggesting that the signal detected in the 25-Clb2 myc tagged strain is specific. Nevertheless, some background is indeed present, possibly explaining part of the unexpected Clb2 protein signal detected during G1.

3. Based on the extensive literature of Clb2-CDK driving mitotic entry and progression, there is a clear prediction that lower Clb2 protein will slow G2/M progression resulting in larger cells at the time of cell division. This is supported by the data in Fig 5 where the ZIP code mutants with lower Clb2 protein levels show larger cells at division. However, this does not explain why mother cells would be larger at the time of budding (Fig 5C), which is controlled by the G1 size control mechanism and at a time when no Clb2 is present. This also doesn't explain why more bud volume is added in G2/M without an increase in budded timing. Is it possible these cells are larger at all cell stages for other reasons?

We agree with the reviewer that some data in the initial submission were confounding. We believe that this was caused by the fact that the temporal resolution of our timelapse imaging was not sufficiently to precisely measure the changes we wanted to capture. While the durations of the budded phase we included in the initial submission did not show significant differences between mutants, those differences (ca. 5 min) were based on datasets in which frames were acquired every 5 minutes. We reacquired images with a higher frame rate (every 60 s) to test if in the data submitted initially the acquisition rate masked differences in duration. We found that imaging GFP once per minute led to slower

growth even at the very low LED power and exposure used in the experiment. Therefore, we repeated the experiment while only acquiring bright field and red fluorescence, which did not reduce growth rate appreciably, and which allowed us to determine the duration of the budded phase with higher accuracy than in the initial submission. We found a highly significant increase in G2/M of around 5 min, consistent with the low time resolution data in the initial submission. Given that a 5 min absolute increase in budded phase duration corresponds to a relative increase in duration of around 10%, the difference in added bud volume in **Fig. 5g** in the initial submission can be explained. Nonetheless, in this resubmission we have omitted subplots **Fig. 5c-g** of the initial submission for clarity.

4. The authors try to link CLB2 mRNA localization and translation efficiency to cell cycle progression and a potential bud sizer mechanism. The crucial measurement is described in the text as “While nuclear fluorescence tightly predicted bud size in WT and the Δshe mutants’ (Fig. 5k and Extended Data Fig. 7m) this effect was significantly more disperse in the ZIP-mutant strain, indicating a loss of communication between the two compartments.” A scatter plot of nuclear fluorescence vs bud size for the two populations would be more appropriate here. The authors also note that the ZIP code mutant added more material during the budded phase than WT (Fig 5g) and spent longer median time in budded phase (Fig 5h, albeit the authors recognize this result is non-significant). The authors conclude that “control of CLB2 mRNA translation in the bud compartment reports bud size”. These two statements (and the overall description of Fig 5) are difficult to parse, and therefore the intended conclusions and interpretation from the data in Fig 5 are unclear. The results need

to be rewritten for clarity, and further experiments are needed that more directly test associations between CLB2 mRNA localization and bud growth. If the intended conclusion is that Clb2 protein levels are regulated by bud size, it would be much more convincing to include other experiments manipulating bud size (latrunculin, or mutants in polarity, vesicle delivery, or septins) and determining the effect on Clb2 protein levels.

As detailed in the point above, we reacquired timelapse data to determine G2/M duration more precisely. We found small but highly significant increases in budded phase duration in all mutants studied (**New Figure 5d,e**). Additionally, we have reacquired microscopy images to determine the correlation of bud compartment size and nuclear fluorescence (**New Figure 5f,g**). To get more reliable signal for both the bud neck marker and the fluorescently labeled Clb2 protein, we acquired images at maximum LED power. To avoid irradiation-induced growth phenotypes, we cultivated agarose pads in the dark and ran a single measurement per position, reasoning that acquiring many snapshot images would preserve correlations between bud compartment size and nuclear fluorescence. In WT, we found strong correlation between nuclear fluorescence and bud size before mitosis (Fig. 5f), which disappeared during and after mitosis (Extended data Fig. 7c). No strong correlation was evident for the ZIP mutant. We proceeded to restructure the content of Fig. 5 and extended data Fig. 7 based on the new data collected and rewrote the relevant sections in the text.

Minor comments

5. In Fig 2 e, f, explicitly state which stage of the cell cycle is being imaged.

Thank you for the suggestion. Cells shown in Figure 2e are in G2 phase. The cells analyzed in Figure 2f include all budded cells. We added this information in Figure 2e and in the legend of Figure 2.

6. The authors assume the ZIP code mutant abrogates She2 binding, but it would be much stronger to show this directly by co-IP.

We agree with the reviewer that with our work we do not test directly the binding of She2 to the *CLB2* ZIP code. However, earlier work from Shepard et al. 2003 independently identified the *CLB2* mRNA as an interacting partner of She2, She3 and Myo4 (PMID: 13679573). Based on those observations and on the new double mutant strains generated for the revised manuscript (*ZIP mut + Δshe2* and *ZIP mut + Δshe3*, **New Extended data figure 5 c,d**), we suggested that a likely hypothesis is that the ZIP code mutant abrogates She2 binding. In the future, we are planning to continue the biochemical characterization of the ZIP code-She2 binding, including by doing co-IP and in-vitro binding experiments.

7. A more careful description of strain construction is needed. Are the 5-myc Clb2 WT and ZIP code mutants constructed at the endogenous locus and in the same way?

The 5-myc and the ZIP code mutants were all integrated at the *CLB2* endogenous locus, and they were generated using the same strategy. Following the reviewer's suggestion, we now extended the Material and methods section describing in more detail plasmids and strains construction.

8. According to the morphogenesis checkpoint model, the presence of a bud is read out by a change in septin organization, which then leads to inactivation of Swe1, the Clb2-CDK inhibitor (Howell and Lew, 2012). A discussion of how the proposed regulation of Clb2 translation relates to the regulation of Clb2 protein activity by the morphogenesis checkpoint would be very useful for readers to synthesize the conclusions here.

We agree with the reviewer that it is important to try to reconcile our results with previous literature. We briefly discussed this point in the Discussion and pointed the readers to relevant literature.

9. In an effort to make scientific language more inclusive, the authors could consider changing references of mother and daughter cells to parent (or originator) and progeny cells (White, et al, 2021, doi: 10.1038/d41586-021-03490-7).

We thank the reviewer for the suggestion. We will incorporate this change in future manuscripts.

Reviewer #2 (Remarks on code availability):

I was able to access the raw data and code in the zenodo repository, but there were no README files that I could find. I'm not very familiar with zenodo, so maybe I just missed it.

We thank the reviewer for noticing that we missed a readme file. We now included a readme file in the new Zenodo repository associated with the revised manuscript.

Reviewer #3 (Remarks to the Author):

This manuscript uses a combination of fixed cell and live cell imaging to explore how localization of the CLB2 mRNA affects protein expression, protein localization, cell size, and cell cycle progression. Overall, the experiments are well done, well described, and the conclusions are supported by the data presented. Much of the paper is a beautiful example of how rigorous quantification can provide new insights into biology. The paper doesn't have all the answers and, in some respects, raises more questions than it answers. It reveals that there must be multiple mechanisms that regulate where translation takes place because translation regulation for CLB2 is very different than ASH1, another well-studied mRNA. The work is exciting in identifying a zip code in the mRNA that is important for localization and perhaps translation.

We would like to thank the reviewer for the positive assessment of our work and for the constructive criticism. We agree with the reviewer that this work (including the new experiments done for the revisions of this manuscript) opens many avenues that we hope we and others will be able to follow up in the future.

One thing I am trying to understand is: the localization mutants only show a subtle decrease in protein expression. Why is localization of the mRNA to the bud important? The authors suggest that the mRNA may be more efficiently translated in the bud than in the mother based on data in Figure 4 that show there is greater colocalization of mRNA and protein in the bud than the mother. But the difference is quite subtle (a mean of less than 1 colocalization spot in the mother compared to ~ 2 colocalization spots in the bud). While the difference is statistically significant it seems extremely small from a biological perspective. Could the authors comment on this observation?

The data in **Fig 4c** are not corrected for compartment size. Rather than concentrations, actual counts are given. The bud compartment is always strictly smaller than the mother compartment. Assuming a bud that has half of the volume of the mother, a 2:1 difference in translation foci becomes a 4:1 difference. At the low counts of mRNAs, we are observing

(<10 at peak expression), 0-1 transcripts located in the mother cell is the smallest amount we can hope for in the presence of localization noise. Biologically, this mechanism is part of a larger mechanism that finetunes Clb2 expression levels. Clb2CDK is subject to post-translational modifications via phosphorylation by e.g. SWE1, which titrates activity.

In Figure 4, the authors show that protein expression increases in G2/M and protein is more frequently found in the bud (but it is still found in the mother more frequently). mRNA expression increases significantly in the bud in G2/M. In WT cells only 25% of mRNA associated with protein suggesting that mRNA is infrequently translated. In the mRNA localization mutant, the authors found more colocalizing mRNA and protein in the mother and less in bud - was this done as a function of the cell cycle (like in WT)? Did authors quantify mRNA and protein (like in c and d) and do they see a decrease in amount of protein at the single cell level (like in WB)?

We thank the reviewer for this suggestion. We did indeed quantify the *CLB2* mRNA and protein in the localization mutant as a function of cell cycle phase, but we did not report the analysis in the first version of the manuscript. Below, on the left, we report Clb2 protein expression measured by immunofluorescence in WT cells and in the $\Delta she2$ mutant. The analysis shows a consistent reduction in Clb2 protein expression across the cell cycle in the localization mutant. This is also visible when the data from the different cell cycle phases is compiled; Figure below, left panel (not included in the manuscript) and right panel, **New Extended Data Figure 6f**. Conversely, *CLB2* mRNA expression does not show differences in the localization mutant, regardless of the cell cycle phase (not shown).

The data presented in Extended Data Figure 7 are confounding. Up until this point, the data in the manuscript seemed to support the argument that the zip code in the mRNA interacts with the she2-she3 system to localize to the bud and localization to the bud affects protein translation and hence protein expression levels (to some extent). But the extended data in figure 7 suggest that delta-she mutants that prevent localization of the mRNA to the bud don't exhibit a cell growth defect. In the discussion, the authors rationalize this observation with the explanation that perhaps the zip code serves a dual function – promoting bud localization and efficient translation. If that is the case, wouldn't we expect to see differences between the delta-she2 mutant and the zip code mutant in the single cell single molecule experiments in Figure 4? As far as I can tell, these experiments weren't done with the zip code mutant. But doing so, could potentially provide experimental evidence for the authors' claims.

We agree with the reviewer that the expectation would be that in the ZIP code mutant we would observe reduced co-localization of protein foci with mRNA by FISH-IF. However, we decided not to do this experiment for this rebuttal for the following reasons. First, as the Clb2 protein expression in the ZIP mutant is very low (20-30% of WT), we expect that the signal would be too noisy to be reliably detect by IF. Considering the effort it would have taken to make the strain and to perform and analyze FISH-IF data, we decided to prioritize other type of experiments we could do with the current lab personnel. Second, our data already demonstrated, although indirectly, that the ZIP code mutant has a translation defect as the protein reduction is not due to a change in mRNA expression, nor protein stability. We remain open to generate this strain for future studies.

Reviewer #3 (Remarks on code availability):

The zenodo link provides all raw data for the paper. It also provides code (in R and Python) for data analysis of FACS data and imaging data. As far as I can tell scripts are provided but there are no readme files. I didn't install and run the code.

We thank the reviewer for noticing that a readme file was missing. We now included the file in the new Zenodo repository associated with the revised manuscript.

Reviewer #4 (Remarks to the Author):

We thank this reviewer for the constructive criticism!

REVIEWERS' COMMENTS

Reviewer #1 (Remarks to the Author):

The authors have properly answered the comments of the reviewers. Overall, this is a very interesting work which uses leading-edge quantitative imaging to unveil a new mechanism linking mRNA localized translation to cell growth and cell cycle progression. I think this manuscript is now acceptable for publication in this journal.

We would like to thank this reviewer for the constructive criticism and for taking the time to evaluate our work.

Reviewer #2 (Remarks to the Author):

In this revision, the authors have greatly improved the experiments and clarity of the text. I am convinced that CLB2 mRNA localization in the bud supports efficient translation of Clb2 protein, and, consistent with the field, the authors have provided convincing evidence that Clb2 protein levels affect cell division timing. However, I remain concerned with the interpretations and conclusions surrounding CLB2 mRNA localization to the bud reflecting bud growth. This paper is strong and doesn't require any further experiments, but the interpretations in the results and discussion sections need to be more cautious.

The data is very convincing that Clb2 protein levels correlate with duration of the budded phase and cell division timing. However, the relationship between Clb2 protein levels and bud size are less straightforward. On several occasions, the authors have attempted to explain that Clb2 protein levels are a function of bud size (a couple examples below), an interpretation that is not well supported with the data in this manuscript. An alternative explanation could be that bud size is only a function of time, i.e. more time in budded phase results in bigger buds. Since lower Clb2 protein levels extend the duration of the budded phase, the bud sizes could simply be larger due to lower Clb2 protein levels. While both interpretations are possible, the data presented in the paper does not directly address this issue. As stated in the original review, if the intended conclusion is that Clb2 protein levels are regulated by bud size, it would be much more convincing to include other experiments that manipulate bud size with independent approaches (latrunculin, or mutants in polarity, vesicle delivery, or septins) and determine the effect on Clb2 protein levels. These experiments are probably outside the scope of the current manuscript, so I recommend just that the causal relationship between Clb2 mRNA localization, protein levels, and bud size be restricted to speculation in the Discussion section.

Example statements to reconsider:

- 297 "Taken together, this chain of dependences can function as a mechanism that measures maturity of the bud compartment; active localization of a transcript to the bud compartment may provide preferential access of that transcript to the ribosomes that

are localized in that compartment, and a larger bud compartment harbors more ribosomes. The concentration of the resulting protein should then reflect bud size.” – This argument is unsupported and speculative.

- 310 “If *CLB2* mRNA localization were part of a mechanism that measures bud maturity, mutants with non-localized transcript would be expected to be born larger, as a result of the parent cell spending more time in the budded phase. Indeed, ZIP mutant cells were on average 10.9% larger at birth than wild type cells (Fig. 5h), but no significant effect was evident for the Δ she mutants (Extended data Fig. 7d,e).” – While the data supports a correlation between *CLB2* mRNA localization and cell size, the text here implies causation where *CLB2* mRNA measures bud maturity.

- 430 “Altogether, our data suggest that by coupling *CLB2* mRNA bud localization and protein synthesis, cells coordinate cell growth with cell cycle progression by sensing the bud translation capacity via *CLB2* mRNA transport and localized translation.” – This point about sensing bud translation capacity is highly speculative and not a direct conclusion of the data presented.

Reviewer #2 (Remarks on code availability):

I was able to access the code and the README file.

We would like to thank this reviewer for the significant effort taken to provide valuable feedback and constructive criticism.

We understand the criticism and have revised the manuscript, specifically the Results and Discussion sections, to reduce the implied causality between *CLB2* mRNA localized translation and bud size control. In the final paragraph of the Discussion, our model now states that *CLB2* mRNA localization and nuclear protein levels may either reflect bud size or the duration of budding phase. This clarification acknowledges both possibilities and emphasizes that further experimental work is needed to test these hypotheses.

Reviewer #3 (Remarks to the Author):

I appreciate the authors' due diligence in preparing this revised manuscript. They have satisfactorily addressed all my concerns - either through experiments or explanations. I am supportive of publication.

Reviewer #3 (Remarks on code availability):

The zenodo link provides all data and code for the paper. It now includes a readme file.

We would like to thank this reviewer for the constructive criticism and for taking the time to review our work.

Reviewer #4 (Remarks to the Author):

We would like to thank this reviewer for the constructive feedback and for dedicating time and effort to evaluate our work.